# Predicting individual traits from models of brain dynamics accurately and reliably using the Fisher kernel

**Christine Ahrends[1]\*[†], Mark W Woolrich[2], Diego Vidaurre[1,3]\***

[1]Center of Functionally Integrative Neuroscience, Department of Clinical Medicine, Aarhus University, Aarhus, Denmark; [2]Oxford Centre for Human Brain Activity, Department of Psychiatry, University of Oxford, Oxford, United Kingdom; [3]Department of Psychiatry, University of Oxford, Oxford, United Kingdom

## eLife Assessment

This **important** study combines the use of Fisher Kernels with Hidden Markov models aiming to improve brain-behaviour prediction. The evidence supporting the authors' conclusions is **compelling**, comparing brain-behaviour prediction accuracies across a range of different traits, including out of sample assessment. This work is timely and will be of interest to neuroscientists working on functional connectivity for brain-behaviour association.

**Abstract** Predicting an individual's cognitive traits or clinical condition using brain signals is a central goal in modern neuroscience. This is commonly done using either structural aspects, such as structural connectivity or cortical thickness, or aggregated measures of brain activity that average over time. But these approaches are missing a central aspect of brain function: the unique ways in which an individual's brain activity unfolds over time. One reason why these dynamic patterns are not usually considered is that they have to be described by complex, high-dimensional models; and it is unclear how best to use these models for prediction. We here propose an approach that describes dynamic functional connectivity and amplitude patterns using a Hidden Markov model (HMM) and combines it with the Fisher kernel, which can be used to predict individual traits. The Fisher kernel is constructed from the HMM in a mathematically principled manner, thereby preserving the structure of the underlying model. We show here, in fMRI data, that the HMM-Fisher kernel approach is accurate and reliable. We compare the Fisher kernel to other prediction methods, both time-varying and time-averaged functional connectivity-based models. Our approach leverages information about an individual's time-varying amplitude and functional connectivity for prediction and has broad applications in cognitive neuroscience and personalised medicine.

## Introduction

Observing a person's behaviour over time is how we understand the individual's personality, cognitive traits, or psychiatric condition. The same should apply at the brain level, where we may be able to gain crucial insights by observing the patterns in which brain activity unfolds over time, that is brain dynamics. One way of describing brain dynamics are state-space models, which allow capturing recurring patterns of activity and functional connectivity (FC) across the whole brain. However, although research into brain dynamics has recently gained traction (**Breakspear, 2017**; **Calhoun et al., 2014**; **Fox et al., 2005**; **Hutchison et al., 2013**; **Liégeois et al., 2017**), it is still unclear how best to use this spatiotemporal level of description to characterise subject differences or predict individual traits from

*For correspondence:
christine.ahrends@cfin.au.dk (CA);
dvidaurre@cfin.au.dk (DV)

Present address: [†]Wellcome Centre for Integrative Neuroimaging, John Radcliffe Hospital, University of Oxford, Oxford, United Kingdom

Competing interest: The authors declare that no competing interests exist.

**eLife digest** Watching how people behave over time can provide insights into their personality, mental health, as well as how they think and problem solve. Like behaviour, brain activity patterns constantly change, both at rest and in response to external events. These changes might reveal crucial information about a person that cannot be seen when looking at a single snapshot or an average of brain activity.

It has been difficult for researchers to predict individual traits from the overarching dynamic patterns of brain activity measured using brain scans and other imaging tools. This is due to the patterns being too complex to be analyzed directly. Mathematical models like the Hidden Markov Model can describe dynamic patterns in brain activity, such as how different brain areas' activity and interaction with one another changes over time. To use this type of model to predict individual traits, Ahrends et al. combined it with a machine learning technique known as the Fisher kernel.

Using this combination of techniques to model dynamic patterns of brain activity based on scans from 1,000 resting people allowed the researchers to successfully predict an individual's age and their score in various cognitive tests. This approach was shown to more accurately predict traits than alternative methods.

In the future, researchers may use this new modeling technique to search for markers of disease in dynamic brain activity patterns. For example, this could provide information about the progression of neuropsychiatric diseases over time. It may also help neuroscientists study how dynamic brain activity patterns contribute to individual cognitive performance.

brain signals. One reason why brain dynamics are not usually considered in this context pertains to their representation: They are represented using models of varying complexity that are estimated from modalities such as functional MRI or MEG. Although there exists a variety of methods for estimating time-varying or dynamic FC (*Lurie et al., 2020*), like the commonly used sliding-window approach, there is currently no widely accepted way of using them for prediction problems. This is because these models are usually parametrised by a high number of parameters with complex mathematical relationships between the parameters that reflect the model assumptions. How to leverage these parameters for prediction is currently an open question.

We here propose the Fisher kernel for predicting individual traits from brain dynamics, using information from generative models that do not assume any knowledge of task timings. We focus on models of brain dynamics that capture within-session changes in functional connectivity and amplitude from fMRI scans, in this case acquired during wakeful rest, and how the parameters from these models can be used to predict behavioural variables or traits. In particular, we use the Hidden Markov Model (HMM), which is a probabilistic generative model of time-varying amplitude and functional connectivity (FC) dynamics (*Vidaurre et al., 2017*). HMMs have previously been shown to be able to predict certain complex subject traits, such as fluid intelligence, more accurately than structural or static (time-averaged) FC representations (*Vidaurre et al., 2021*). We combine the HMM with the Fisher kernel, which allows for the efficient use of the entire set of parameters from the generative model. The Fisher kernel takes the complex relationships between the model parameters into account by preserving the structure of the underlying model (here, the HMM; *Jaakkola et al., 1999*; *Jaakkola and Haussler, 1998*). Mathematically, the HMM parameters lie on a Riemannian manifold (the structure). This defines, for instance, the relation between parameters, such as: how changing one parameter, like the probabilities of transitioning from one state to another, would affect the fitting of other parameters, like the states' FC. It also defines the relative importance of each parameter; for example, how a change of 0.1 in the transition probabilities would not be the same as a change of 0.1 in one edge of the states' FC matrices.

For empirical evaluation, we consider two criteria that are important in both scientific and practical applications. First, predictions should be as accurate as possible, that is the correlation between predicted and actual values should be high. Second, predictions should be reliable, in the sense that a predictive model should never produce excessively large errors, and the outcome should be robust to reasonable variations in the data set, for example the choice of which subjects from the same population are included in the training set. The latter criterion is especially important if we want to be

able to meaningfully interpret prediction errors, for example in assessing brain age (*Cole and Franke, 2017*; *Denissen et al., 2022*; *Smith et al., 2019*). Despite this crucial role in interpreting model errors, reliability is not often considered in models predicting individual traits from neuroimaging features.

In summary, we show that using the Fisher kernel approach, which preserves the mathematical structure of the underlying HMM, we can predict individual traits from patterns of brain dynamics accurately and reliably. We show that our approach significantly outperforms methods that do not take the mathematical structure of the model into account, as well as methods based on time-averaged FC that do not consider brain dynamics. For interpretation, we also investigate which aspects of the model drive the prediction accuracy, both in real data and in simulations. Bringing accuracy, reliability and interpretation together, this work opens possibilities for practical applications such as the development of biomarkers and the investigation of individual differences in cognitive traits.

## Results

We here aimed to predict behavioural and demographic variables from a model of brain dynamics using different kernel functions. The general workflow is illustrated in *Figure 1*. We started with the concatenated fMRI time-series of a group of subjects (*Figure 1*, step 1), here the resting-state fMRI timeseries of 1001 subjects from the Human Connectome Project (HCP, described in detail in section 'HCP imaging and behavioural variables'). We estimated a model of brain dynamics, here the Hidden Markov Model (HMM), which is a state-space model of time-varying amplitude and FC. The HMM and its parameters are explained in detail in section 'The Hidden Markov Model'. We estimated the model at the group level, where the state descriptions, initial state probabilities, and the state transition probability matrix are shared across subjects (*Figure 1*, step 2). Next, we estimated subject-specific versions of this group-level model by dual estimation, where the group-level HMM parameters are re-estimated to fit the individual-level timeseries (*Vidaurre et al., 2021*; *Figure 1*, step 3).

Next, we used this (HMM-mediated) description of the individuals' brain dynamics to predict their individual traits. The parameters of the model lie on a Riemannian manifold, which is a space that has some degree of curvature, illustrated by the curved structure in *Figure 1*, step 4. We mapped these parameters into a feature space. This step works in different ways for the different kernels: In the naïve kernel (step 4a), the features are simply the parameters in the Euclidean space (i.e. ignoring the curvature of the space in *Figure 1*, step 4a); while in the Fisher kernel (step 4b), the features are mapped into the gradient space, which is a tangent space to the Riemannian manifold. We then estimated the similarity between each pair of subjects in this feature space using kernel functions. In this way, we can compare kernels that do not take the structure of the underlying model into account (the naïve kernels) with a kernel that preserves this structure (the Fisher kernel). We also compared these kernels to a previously established method based on Kullback-Leibler divergence, which estimates the similarity between the probability distributions of each pair of individual HMMs. The different kernels are described in more detail in section 'Kernels from Hidden Markov models'.

Finally, we used these kernels to predict the behavioural variables using kernel ridge regression (*Figure 1*, step 5, described in detail in section 'Predictive model: Kernel ridge regression'). The first three steps are identical for all kernels and therefore carried out only once. The fourth step (mapping the examples and constructing the kernels) is carried out once for each of the different kernels. The last step is repeated 3500 times for each kernel to predict a set of 35 different behavioural variables using 100 randomised iterations of 10-fold nested cross validation (CV). We evaluated 24,500 predictive models using different kernels constructed from the same model of brain dynamics in terms of their ability to predict phenotypes, as well as another 24,500 predictive models based on time-averaged features, described in detail in section 'Models based on time-averaged FC features'.

### The Fisher kernel predicts more accurately than Euclidean methods

Using the resting-state fMRI timeseries from the HCP dataset, we found that among the kernels constructed from HMMs, the linear Fisher kernel had the highest prediction accuracy on average across the range of behavioural variables and CV folds and iterations, as shown in *Figure 2a*. Compared to the other linear kernels (which do not respect the geometry of the HMM parameters), the linear Fisher kernel (mean $r$ $\kappa_{Fi}$: 0.192) was significantly more accurate than the linear naïve kernel (mean $r$ $\kappa_{Ni}$: 0.05, $t_{rkCV}$=2.631, $p_{BH}$=0.031). The comparison with the linear naïve normalised kernel was not significant

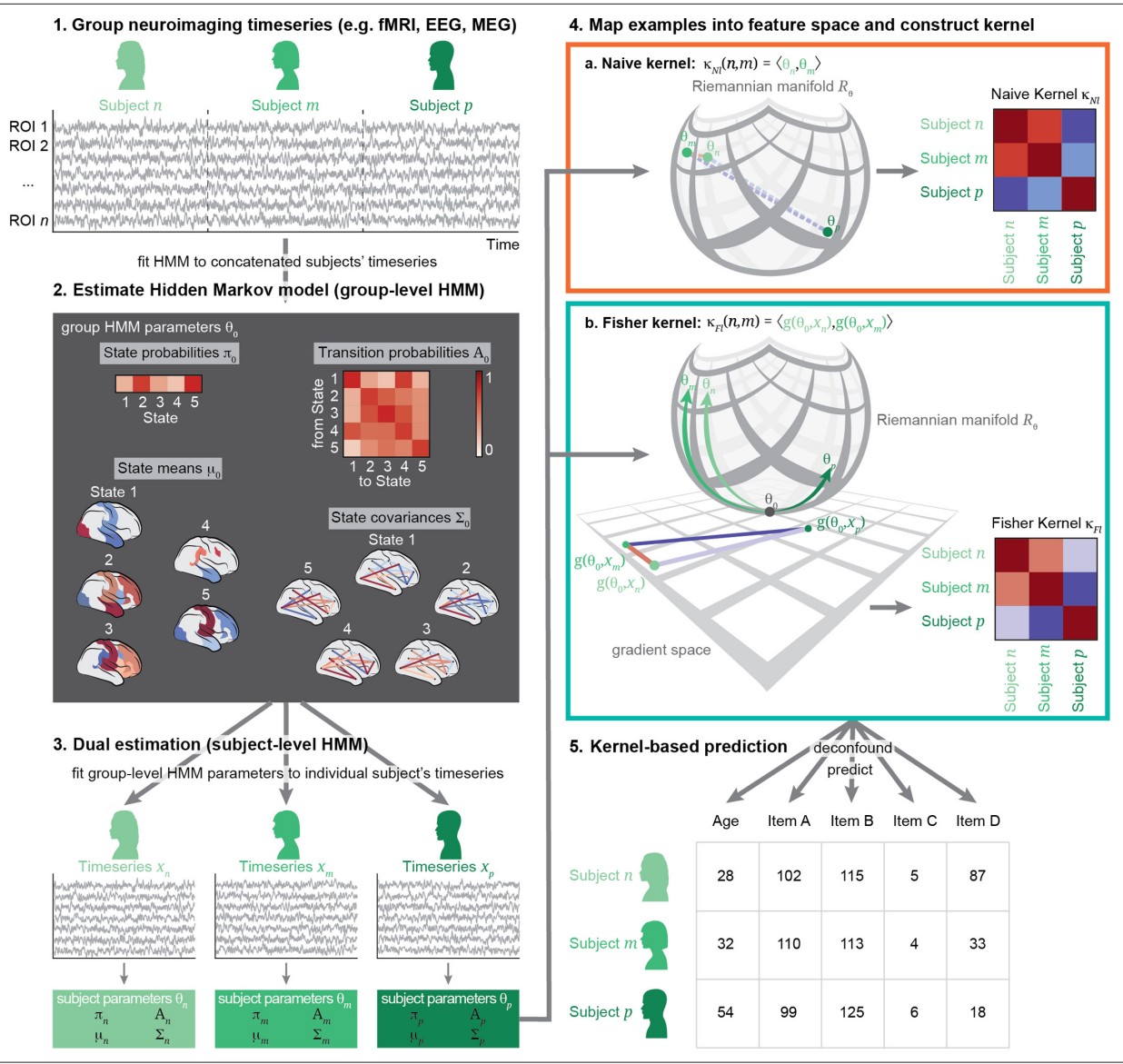

**Figure 1.** Workflow of the Fisher kernel prediction approach. To generate a description of brain dynamics, we (1) concatenate all subjects' individual timeseries; then (2) estimate a Hidden Markov Model (HMM) on these timeseries to generate a group-level model; then (3) dual-estimate into subject-level HMM models. Steps 1–3 are the same for all kernels. In order to then use this description of all subjects' individual patterns of brain dynamics, we map each subject into a feature space (4). This mapping can be done in different ways: In the naïve kernels (4a), the manifold (i.e. the curved structure) on which the parameters lie is ignored and examples are treated as if they were in Euclidean space. The Fisher kernel (4b), on the other hand, respects the structure of the parameters in their original Riemannian manifold by working in the gradient space. We then construct kernel matrices ($\kappa$), where each pair of subjects has a similarity value given their parameters in the respective embedding space. Finally, we feed $\kappa$ to kernel ridge regression to predict a variety of demographic and behavioural traits in a cross-validated fashion (5).

(mean $r$ $\kappa_{NNl}$: 0.153, $t_{rkCV}$=1.022, $p_{BH}$=0.192). This indicates a positive effect of using a tangent space embedding rather than incorrectly treating the HMM parameters as Euclidean, but that this effect can be mitigated by normalising the parameters before constructing the kernel.

Among the Gaussian kernels, the Gaussian Fisher kernel also had the highest average prediction accuracy (mean $r$ $\kappa_{Fg}$: 0.166), though the comparisons with the other kernels were not significant (mean $r$ $\kappa_{Ng}$: 0.107, $t_{rkCV}$=1.277, $p_{BH}$=0.173; mean $r$ $\kappa_{NNg}$: 0.094, $t_{rkCV}$=1.466, $p_{BH}$=0.145; mean $r$ $\kappa_{KL}$: 0.163, $t_{rkCV}$=0.069, $p_{BH}$=0.482). Comparing prediction accuracies of the linear with the Gaussian Fisher kernel was not significant ($t_{rkCV}$=0.993, $p_{BH}$=0.192).

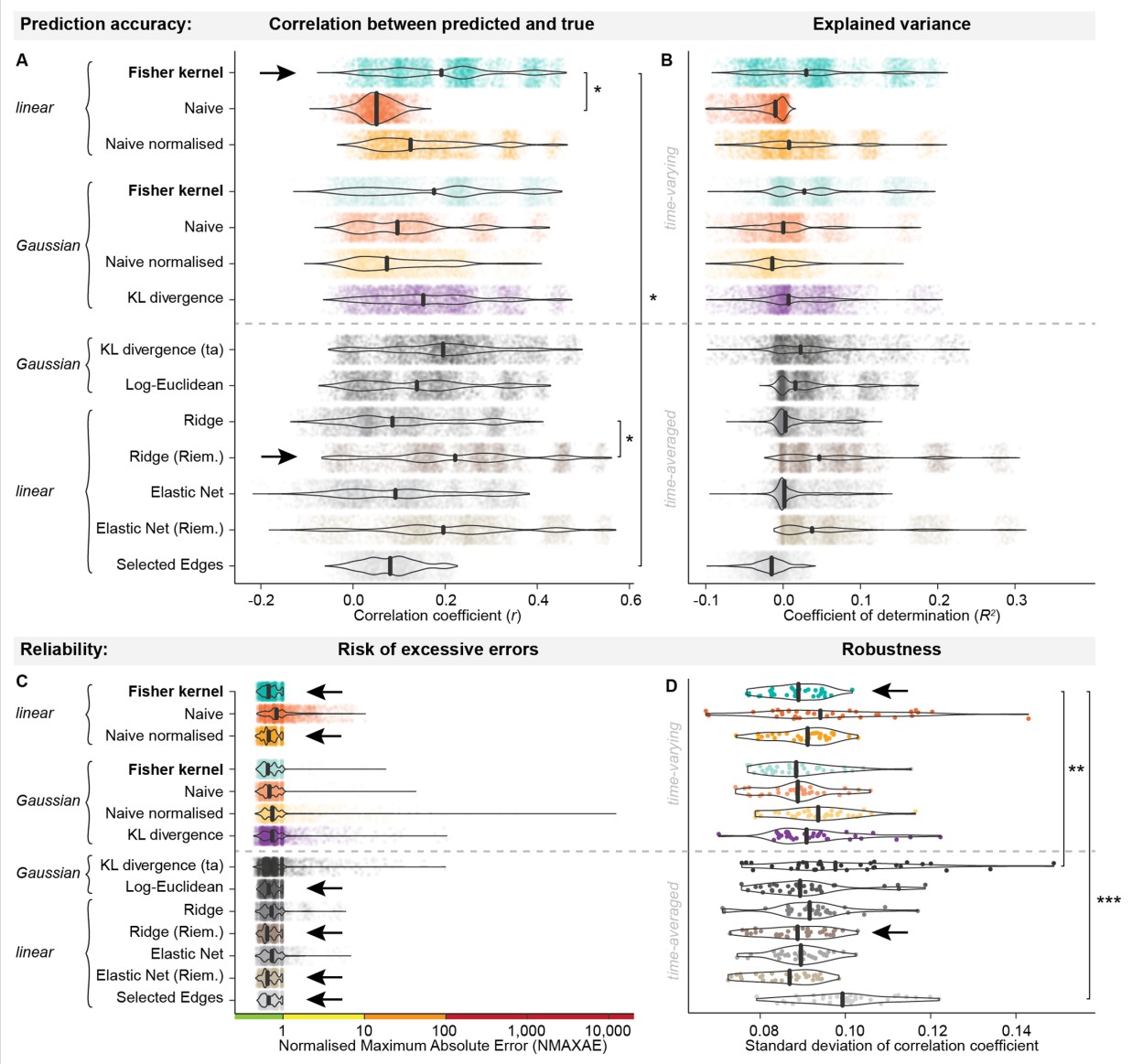

**Figure 2.** Distributions of performance across subject traits and CV iterations when using different methods for prediction of subject traits on HCP data. The best-performing methods are highlighted by black arrows in each plot. (**a**) Pearson's correlation coefficients (r) between predicted and actual variable values in deconfounded space as a measure of prediction accuracy (x-axis) of each method (y-axis). Larger values indicate that the model predicts more accurately. The linear Fisher kernel has the highest average accuracy among the time-varying methods, while the Ridge regression model in Riemannian space had the highest average accuracy among the time-averaged methods. Note that we here show the distribution across target variables and CV iterations but averaged over folds for visualisation purposes, while the fold-wise accuracies were used for significance testing. Asterisks indicate significant Benjamini-Hochberg corrected p-values of repeated k-fold cross-validation corrected t-tests below 0.05 (*). (**b**) Coefficient of determination ($R^2$) in deconfounded space (x-axis) for each of the methods (y-axis). The x-axis is cropped at –0.1 for visualisation purposes since individual runs can produce large negative outliers, see panel c. (**c**) Normalised maximum absolute errors (NMAXAE) in original (non-deconfounded) space as a measure of excessive errors (x-axis) by method (y-axis). Large maximum errors indicate that the model predicts very poorly in single cases. Differences between the methods mainly lie in the tails of the distributions, where the naïve normalised Gaussian kernel produces extreme maximum errors in some runs (NMAXAE >10,000), while the linear naïve normalised kernel and the linear Fisher kernel, along with several time-averaged methods have the smallest risk of excessive errors (NMAXAE below 1). The x-axis is plotted on the log-scale. (**d**) Robustness of prediction accuracies. The plot shows the distribution across variables of the standard deviation of correlation coefficients over folds and CV iterations on the x-axis for each method (on the y-axis). Smaller values indicate greater robustness. The linear Fisher kernel and the time-averaged Ridge regression model in Riemannian space are the most robust. Asterisks indicate significant Benjamini-Hochberg corrected p-values for repeated measures t-tests below 0.01 (**) and 0.001 (***). (**a**, **b**, **c**) Each violin plot shows the distribution over 3500 runs (100 iterations of 10-fold CV for all 35 variables) that were predicted from each method. (**d**) Each violin plot shows the distribution over 35 variables that were predicted from each method.

*Figure 2 continued on next page*

The online version of this article includes the following figure supplement(s) for figure 2:

**Figure supplement 1.** Average errors.

**Figure supplement 2.** Results from using only one scanning session per participant.

**Figure supplement 3.** Results from HMM where states are defined only in terms of covariance and mean is set to 0.

**Figure supplement 4.** Effect of hyperparameters on model errors.

Compared to the methods using time-averaged FC for prediction, the linear Fisher kernel significantly outperformed the Selected Edges method (mean $r$ Selected Edges: 0.081, $t_{rkCV}$=2.417, $p_{BH}$=0.031). The average prediction accuracy of the linear Fisher kernel is also higher than the log-Euclidean kernel (mean $r$: 0.140, $t_{rkCV}$=1.456, $p_{BH}$=0.145) and the Ridge regression model (mean $r$: 0.111, $t_{rkCV}$=1.906, $p_{BH}$=0.085), but outperformed by time-averaged KL divergence (mean $r$: 0.194, $t_{rkCV}$=−0.044, $p_{BH}$=0.482) and Ridge regression in Riemannian space (mean $r$ Ridge Riem.: 0.223, $t_{rkCV}$=−0.999, $p_{BH}$=0.192), though these comparisons were all not significant. Analogous to the effect of tangent space embedding for the HMM parameters (linear Fisher kernel compared to linear naïve kernel), using a tangent space embedding on the time-averaged covariance matrices (Ridge regression in Riemannian space compared to Ridge regression) also significantly improved the prediction accuracy ($t_{rkCV}$=−2.537, $p_{BH}$=0.0313). The Elastic Net and Elastic Net in Riemannian space were not used for statistical comparisons since they failed to converge in a substantial portion of runs (>20%). The Elastic Nets showed similar performance to the Ridge regression models. We also observed

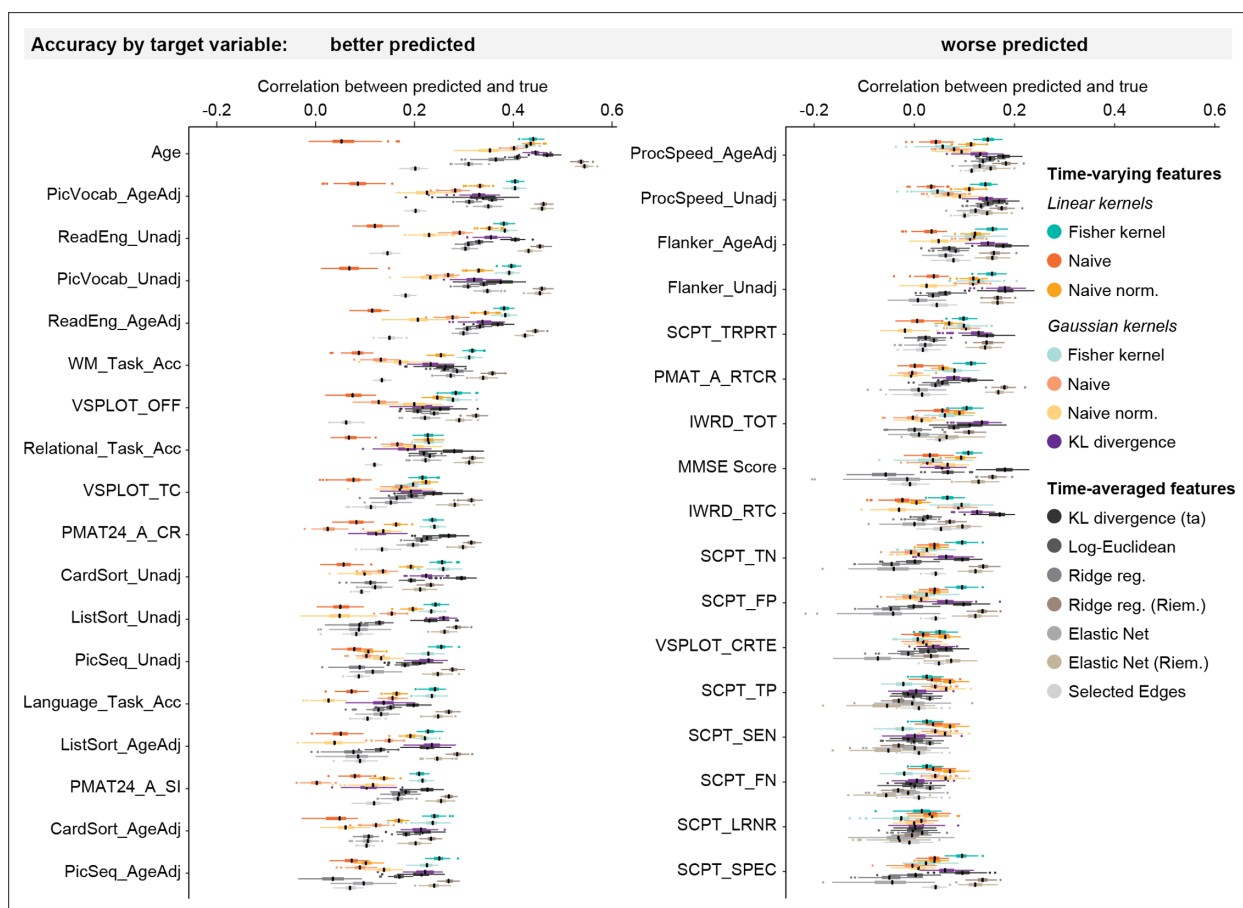

**Figure 3.** Model performance estimates over cross-validation (CV) iterations by behavioural variable and method, ordered by accuracy on HCP data. Boxplots show the distribution over 100 iterations of 10-fold CV of correlation coefficient values (x-axis) of each method, separately for each of the 35 predicted variables (y-axes). Among the time-varying methods, the linear Fisher kernel (green) predicts at higher accuracy for many variables, and also shows the narrowest range, indicating high robustness. However, for many target variables, it is outperformed by the time-averaged tangent space models (Ridge reg. Riem. and Elastic Net Riem.). Black lines within each boxplot represent the median.

that the non-kernel-based Euclidean time-averaged FC models, that is the Ridge regression and the Selected Edges model, make predictions at a smaller range than the actual variables, close to the variables' means. This leads to weak relationships between predicted and actual variables but smaller errors, as shown in *Figure 2—figure supplement 1*. The distributions of correlation coefficients for the different methods are shown in *Figure 2a* and the explained variance in *Figure 2b*. The performance of all methods is also summarised in *Supplementary file 1b*.

We found that the linear Fisher kernel also predicted more accurately than other HMM-based kernels when fitting the HMM to only the first resting-state session of each subject, as shown in *Figure 2—figure supplement 2*. However, the overall accuracies of all kernels were lower in this case, indicating that predicting traits benefits from a large amount of available data per subject. Similarly, the Fisher kernel outperformed the other kernels when HMM states were defined only in terms of covariance (not mean) at comparable accuracy, as shown in *Figure 2—figure supplement 3*.

As shown in *Figure 3*, there are differences in how well these demographic or behavioural variables can be predicted from a model of brain dynamics. Certain variables may be more related to static or structural measures (*Liégeois et al., 2019*; *Vidaurre et al., 2021*), or just be difficult to predict in general. Overall, age could be best predicted, followed by language-related cognitive items (PicVocab, ReadEng both age adjusted and unadjusted).

In summary, the linear Fisher kernel has the highest prediction accuracy of the time-varying methods, significantly outperforming the linear naïve kernel which does not take the geometry of the HMM parameters into account. The linear Fisher kernel also has a higher prediction accuracy than several methods using time-averaged FC features, but it is outperformed by time-averaged methods that work in tangent space.

## The linear Fisher kernel has a lower risk of excessive errors and is more robust than other methods

We now show empirically that the linear Fisher kernel is more reliable than other kernels, both in terms of risk of large errors and in terms of robustness over CV iterations.

The linear versions of the Fisher kernel and the naïve normalised kernel had the overall lowest risk of large errors among the time-varying methods, as shown in *Figure 2c*. We assessed the risk of large errors (NMAXAE >10), very large errors (NMAXAE >100), and extreme errors (NMAXAE >1,000), corresponding to one, two, and three orders of magnitude of the range of the actual variable. For the Fisher kernel, the risk of large errors is low: 0% in the linear version $\kappa_{Fl}$ and 0.029% in the Gaussian version $\kappa_{Fg}$. That means that the linear Fisher kernel never makes large errors exceeding the range of the actual variable by orders of magnitude. In the naïve kernel, both the linear $\kappa_{Nl}$ and the Gaussian version $\kappa_{Ng}$ have a low risk of large errors at 0.057% for the linear version and 0.029% for the Gaussian version. While the linear naïve normalised kernel $\kappa_{NNl}$ has a 0% risk of large errors, its Gaussian version $\kappa_{NNg}$ has the overall highest risk of large errors at 1.229%, a risk of very large errors at 0.143%, and even a risk of extreme errors at 0.029%. The KL divergence model $\kappa_{KL}$ has a 0.686% risk of large errors and a 0.029% risk of very large errors. The time-averaged KL divergence model performs slightly better than the time-varying KL divergence, but also has a risk of large errors at 0.600%. The other time-averaged models had no risk of excessive errors. The maximum error distributions are shown in *Figure 2c*.

A reason for the higher risk of large errors in the Gaussian kernels is likely that the radius $\tau$ of the radial basis function needs to be selected (using cross-validation), introducing an additional factor of variability and leaving more room for error. *Figure 2—figure supplement 4* shows the relation between the estimated hyperparameters (the regularisation parameter $\lambda$ and the radius $\tau$ of the radial basis function) and how large errors in the predictions may be related to poor estimation of these parameters.

With respect to robustness, we found that the linear Fisher kernel $\kappa_{Fl}$ had the most robust performance among the time-varying methods, and the Ridge regression model in Riemannian space among the time-averaged methods, on average across the range of variables tested, as shown in *Figure 2d*. Robustness was quantified as the standard deviation of the correlation between model-predicted and actual values over 100 iterations and 10 folds of CV. A low standard deviation indicates high robustness since the method's performance does not differ greatly depending on the specific subjects it was trained and tested on.

Among the time-varying methods, the linear Fisher kernel was the most robust method ($\kappa_{FI}$ mean S.D. $r$: 0.088), though the comparison with the other linear kernels was not significant (linear naïve kernel $\kappa_{NI}$ mean S.D. $r$: 0.096, $t=-2.351$, $p_{BH}=0.099$; linear naïve normalised kernel $\kappa_{NNI}$ mean S.D. $r$: 0.090, $t=-1.745$, $p_{BH}=0.180$). Similarly, the comparison between the Gaussian Fisher kernel ($\kappa_{Fg}$ mean S.D. $r$: 0.089) and the other Gaussian kernels was not significant (Gaussian naïve kernel $\kappa_{Ng}$ mean S.D. $r$: 0.089, $t=0.311$, $p_{BH}=0.826$; Gaussian naïve normalised kernel $\kappa_{NNg}$ mean S.D. $r$: 0.093, $t=-1.802$, $p_{BH}=0.180$; KL divergence $\kappa_{KL}$ mean S.D. $r$: 0.093, $t=-1.758$, $p_{BH}=0.180$). Compared to the time-averaged methods, the linear Fisher kernel was significantly more robust than the time-averaged KL divergence model (mean S.D. $r$: 0.100, $t=-3.798$, $p_{BH}=0.003$) and the Selected Edges model (mean S.D. $r$: 0.100, $t=-5.446$, $p_{BH} <0.0001$), while the other comparisons were not significant (log-Euclidean mean S.D. $r$: 0.091, $t=-1.317$, $p_{BH}=0.277$; Ridge reg. mean S.D. $r$: 0.091, $t=-1.451$, $p_{BH}=0.268$; Ridge reg. Riem. mean S.D. $r$: 0.088, $t=0.059$, $p_{BH}=0.953$). This large variation in model performance depending on the CV fold structure in the time-averaged KL divergence model and the Selected Edges method is problematic. Among the time-averaged methods, the Ridge Regression model in Riemannian space was the most robust method. The ranges in model performance across CV iterations for each variable of the different kernels are shown in *Figure 3*.

Overall, the linear Fisher kernel was the most reliable method among the time-varying methods, and the Ridge regression model in Riemannian space was the most reliable among the time-averaged methods. For both methods, there was no risk of large errors and the variability over CV iterations was the smallest. The Gaussian kernels had higher risks of large errors, and both the time-varying and the time-averaged KL divergence model risked producing large errors, indicating that their performance was less reliable. The Gaussian naïve normalised kernel was the most problematic in terms of reliability with a risk of extreme errors ranging up to four orders of magnitude of the actual variable's range. Two of the time-averaged methods, the time-averaged KL divergence model and the Selected Edges method, showed problems with robustness, indicating considerable susceptibility to changes in CV folds.

## State features drive predictions of individual differences for Fisher kernel

To understand which features drive the prediction, we next simulated timeseries of two groups of subjects that were different either in the mean amplitude of one state or in the transition probabilities. As shown in *Figure 4a*, when we simulated two groups of subjects that are different in terms of the mean amplitude, the Fisher kernel was able to recover this difference in all runs with 0% error, meaning that it identified all subjects correctly in all runs. The Fisher kernel significantly outperformed the other two kernels (Fisher kernel $\kappa_{FI}$ vs. naïve kernel $\kappa_{NI}$: p=0.0003, vs. naïve normalised kernel $\kappa_{NNI}$: p=0.0001). There was no significant difference between the naïve and the naïve normalised kernel (p=1). However, when we simulated differences in transition probabilities between two groups, neither of the kernels were able to reliably recover this difference. In this simulation, the Fisher kernel performed significantly worse than the other two kernels on average (compared to naïve kernel: p=0.006, compared to naïve normalised kernel: p=0.004), as shown in *Figure 4b*. As in the previous simulation, the naïve kernel and the naïve normalised kernel did not significantly differ from each other (p=1).

The above results suggest that all kernels, but particularly the Fisher kernel, are most sensitive to differences in state parameters rather than differences in transition probabilities. To understand whether the difference in transition probabilities can be recovered when it is not overshadowed by the more dominant state parameters, we ran the second case of simulations again, where we introduced a group difference in terms of transition probabilities, but this time we exclude the state parameters when we constructed the kernels. As shown in *Figure 4c*, the Fisher kernel was now able to recover the group difference with minimal errors, while the naïve normalised kernel improved but did not perform as well as the Fisher kernel. The naïve kernel performed below chance. The Fisher kernel significantly outperformed both the naïve kernel (p=0.0001) and the naïve normalised kernel (p=0.02), and the naïve normalised kernel was significantly more accurate than the naïve kernel (p=0.0004). This shows that the Fisher kernel can recover the group difference in transition probabilities when this difference is not overshadowed by the state parameters. The features and kernel matrices of example runs for all three simulations are shown in *Figure 4a–c* middle and right panels.

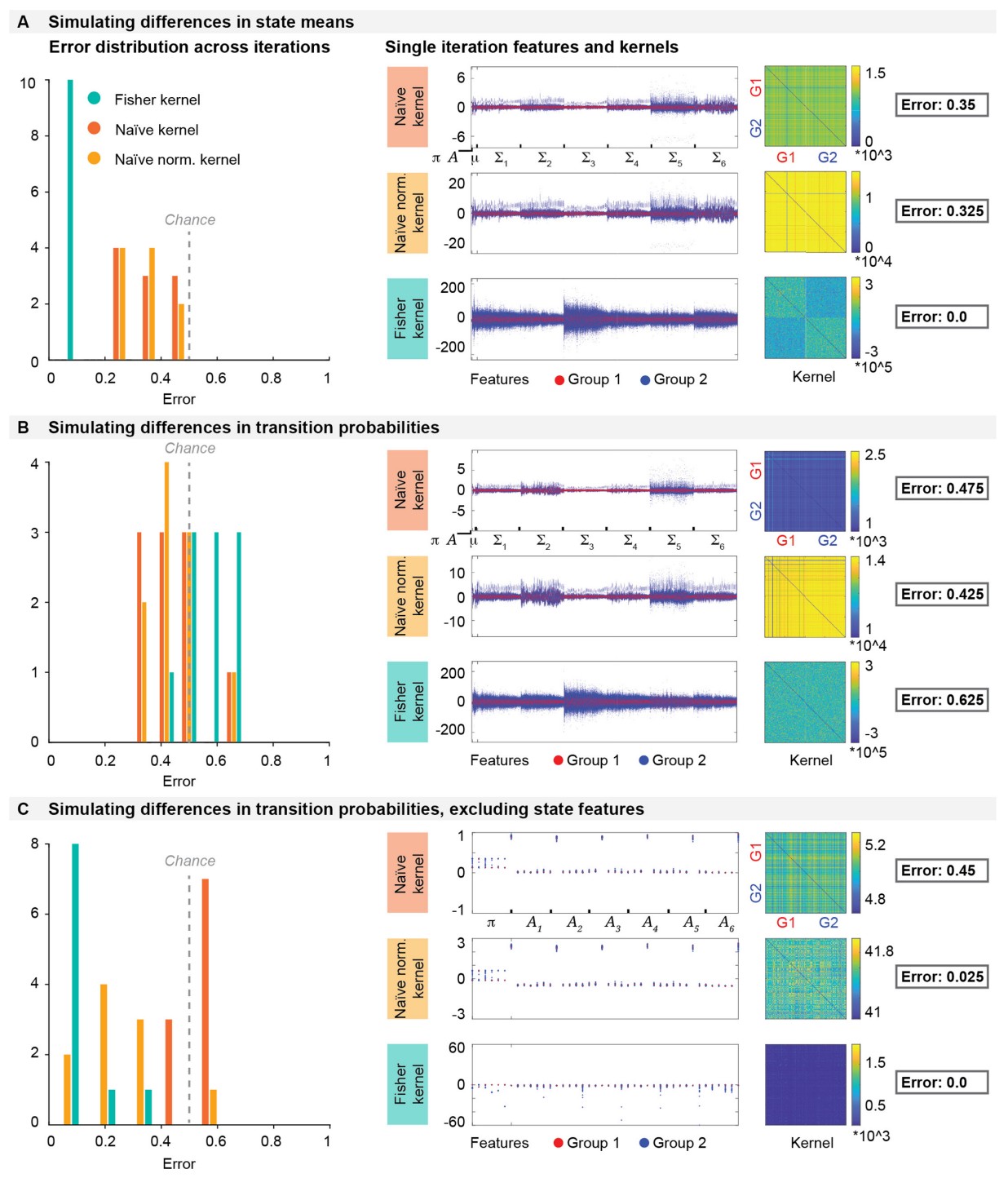

**Figure 4.** Simulations. (**a**) Simulating two groups of subjects that are different in their state means. The error distributions of all 10 iterations show that the Fisher kernel recovers the simulated group difference in all runs with 0% error. (**b**) Simulating two groups of subjects that are different in their transition probabilities. Neither kernel is able to reliably recover the group difference in all 10 iterations. (**c**) Simulating two groups of subjects that are different in their transition probabilities but excluding state parameters when constructing the kernels. The Fisher kernel performs best in recovering the group difference.

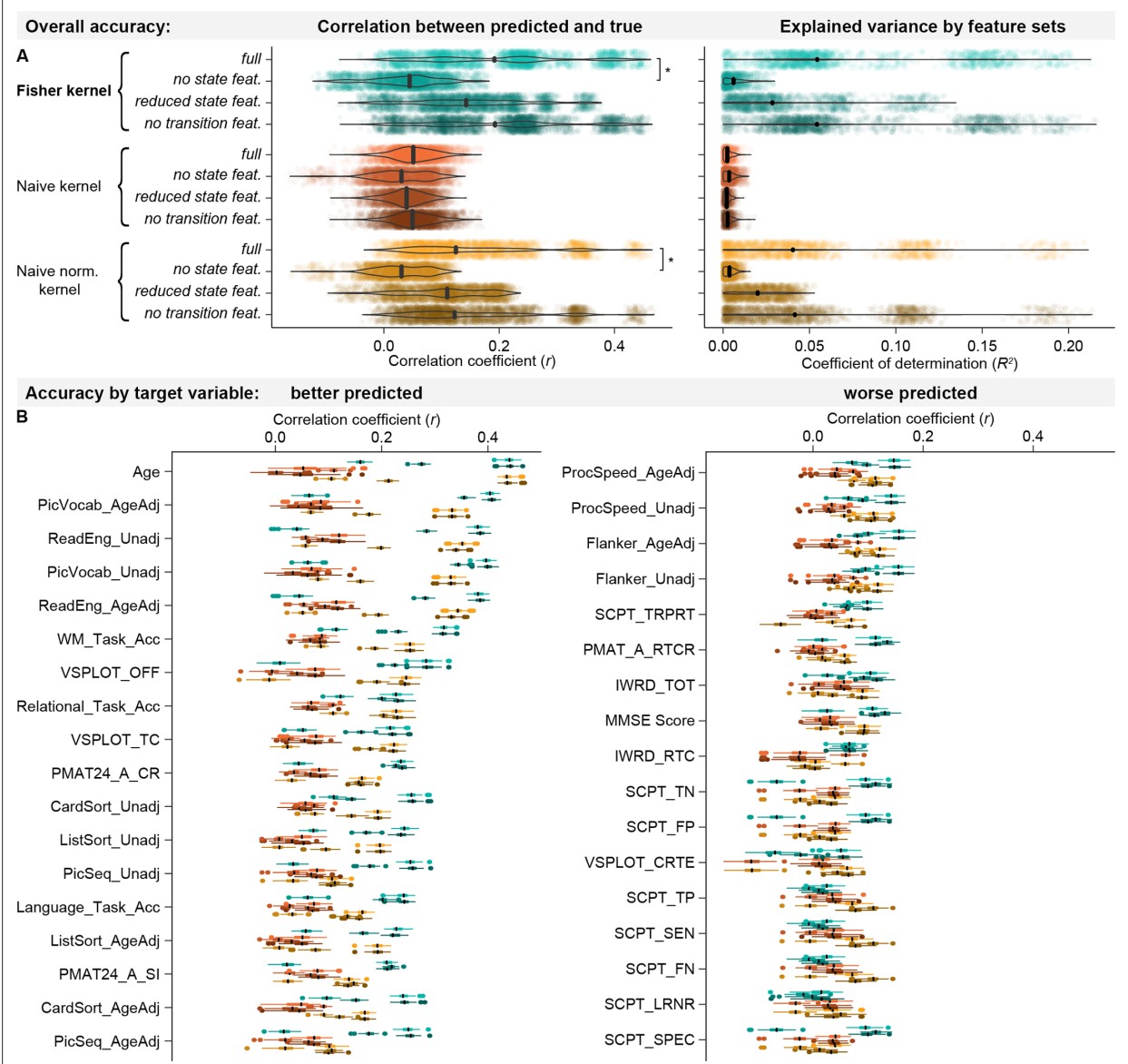

**Figure 5.** Effects of removing sets of features from the kernels on prediction accuracies. (**a**) In the overall prediction accuracies, removing state features significantly decreased performance in the Fisher kernel and the naïve normalised kernel, while removing transition features had no significant effect. (**b**) Removing features has similar effects on all variables, both better predicted (left panel) and worse predicted ones (right panel).

When using real data, the features driving the prediction may differ from trait to trait: For some traits, state parameters may be more relevant, while for other traits, transitions may be more relevant. Given our above findings on simulated data, we therefore compared the effects of systematically removing state features or removing transition features in all traits in the real data.

We found that state parameters were the most relevant features for the Fisher kernel predictions in all traits: As shown in **Figure 5a**, the prediction accuracy of the Fisher kernel was significantly diminished when state features were removed ($t_{rkCV}$ = 2.922, $p_{BH}$ = 0.016), while removing transition features had no significant effect ($t_{rkCV}$ = –0.173, $p_{BH}$ = 0.480). We observed the same effect in the naïve normalised kernel (no state features: $t_{rkCV}$ = 2.460, $p_{BH}$ = 0.031; no transition features: $t_{rkCV}$ = 0.264, $p_{BH}$ = 0.480). For the naïve kernel, removing features did not have any significant effects (no state features: $t_{rkCV}$ = 0.501, $p_{BH}$ = 0.462; no transition features: $t_{rkCV}$=0.050, $p_{BH}$ = 0.480). One reason for the dominance of state parameters may simply be that the state parameters outnumber the other parameters: In the full kernels, we have 15,300 state features (300 features associated with the state means

and 15,000 features associated with the state covariances), but only 42 transition features (6 features associated with the state probabilities and 36 features associated with the transition probabilities). To compensate for this imbalance, we also constructed a version of the kernels where state parameters were reduced to the same amount as transition features using Principal Component Analysis (PCA), so that we have 84 features in total (42 transition features and the first 42 PCs of the 15,300 state features). These PCA-kernels performed better than the ones where state features were removed, but worse than the kernels including all features at the original dimensionality, although not significantly (Fisher kernel: $t_{rkCV}$ = 1.434, $p_{BH}$ = 0.170; naïve kernel: $t_{rkCV}$ = 0.860, $p_{BH}$ = 0.351; naïve normalised kernel: $t_{rkCV}$ = 1.555, $p_{BH}$ = 0.170). This indicates that the fact that state parameters are more numerous than transition parameters does not explain why kernels including state features performed better. Instead, the content of the state features is more relevant for the prediction than the content of the transition features.

When looking at the performance separately for each variable (*Figure 5b*), we found that all variables were better predicted by the version of the kernel which included state features than the ones where state features were removed, while it did not seem to matter whether transition features were included. This indicates that the simulation case described above, where the relevant changes are in the transition probabilities, did not occur in the real data. In certain variables, reducing state features using PCA improved the accuracy compared to the full kernels. This is not unexpected since feature dimensionality reduction is known to be able to improve prediction accuracy by removing redundant features (*Mwangi et al., 2014*).

Empirically, we thus found that the Fisher kernel is most sensitive to individual differences in state parameters, both in simulated timeseries and in the real data. This means that predictions are driven more by what an individual's states look like, rather than by how they transition between states. However, the Fisher kernel can be modified to recover differences in transition probabilities if these are relevant for specific traits.

## Separation between training and test set in HMM training

In the results above, we have constructed kernels from HMMs fit to all subjects in the dataset and only separated them into training and test set at the regression (and deconfounding) step. In machine learning, the gold standard is considered to be a full separation between training and test set at all stages of (pre-)processing to avoid data leakage from the test set into the training set (*Kapoor and Narayanan, 2023*; *Poldrack et al., 2020*). However, a recent study in neuroimaging has found minimal effects of data leakage for breaches of separation between training and test set not involving the target variable or feature selection (*Rosenblatt et al., 2024*). To test whether training the HMM on all subjects may have inflated prediction accuracies, we repeated the HMM-based predictions for kernels that were constructed from HMMs trained only on training subjects. Consistent with the results in *Vidaurre et al., 2021*, for all kernels, both linear and Gaussian versions, the separation of training and test subjects before vs. after fitting the HMM had no significant effects on prediction accuracies (training together vs. separate: linear Fisher kernel: $t_{kCV}$ = 0, $p_{BH}$ = 1; Gaussian Fisher kernel: $t_{kCV}$ = 0, $p_{BH}$ = 1; linear naïve kernel: $t_{kCV}$ = −0.139, $p_{BH}$ = 1; Gaussian naïve kernel: $t_{kCV}$ = −0.018, $p_{BH}$ = 1; linear naïve normalised kernel: $t_{kCV}$ = −0.030, $p_{BH}$ = 1; Gaussian naïve normalised kernel: $t_{kCV}$ = 0.040, $p_{BH}$ = 1). *Figure 6a* shows the distribution of accuracies across target variables and CV folds for all kernels and training schemes, confirming that at this sample size (N=1001), the effect is negligible.

Importantly, in the Fisher kernel, individual subjects' features are defined in reference to a group 'average'. While this has no effect in situations where training and test subjects are taken from the same distribution, it may introduce a bias where test subjects are taken from a different distribution (e.g. the case where one might want to train a model on healthy controls and test on patients). To illustrate this effect, we simulated timeseries for two groups of subjects with varying degrees of between-group difference and noise on the target variable. As shown in *Figure 6b*, this does not affect the kernels when training the HMM on all subjects (training scheme: together, middle panel), but the performance of the Fisher kernel worsens as group difference increases when training the HMM only on the training set (training scheme: separate, right panel). This is because in the Fisher kernel, subjects' similarity is determined by their difference from the group-level HMM. When the test subjects are included in this group-level HMM, their scores will be similar to the training subjects' scores, but when they are excluded from the group-level HMM, their scores may be overestimated

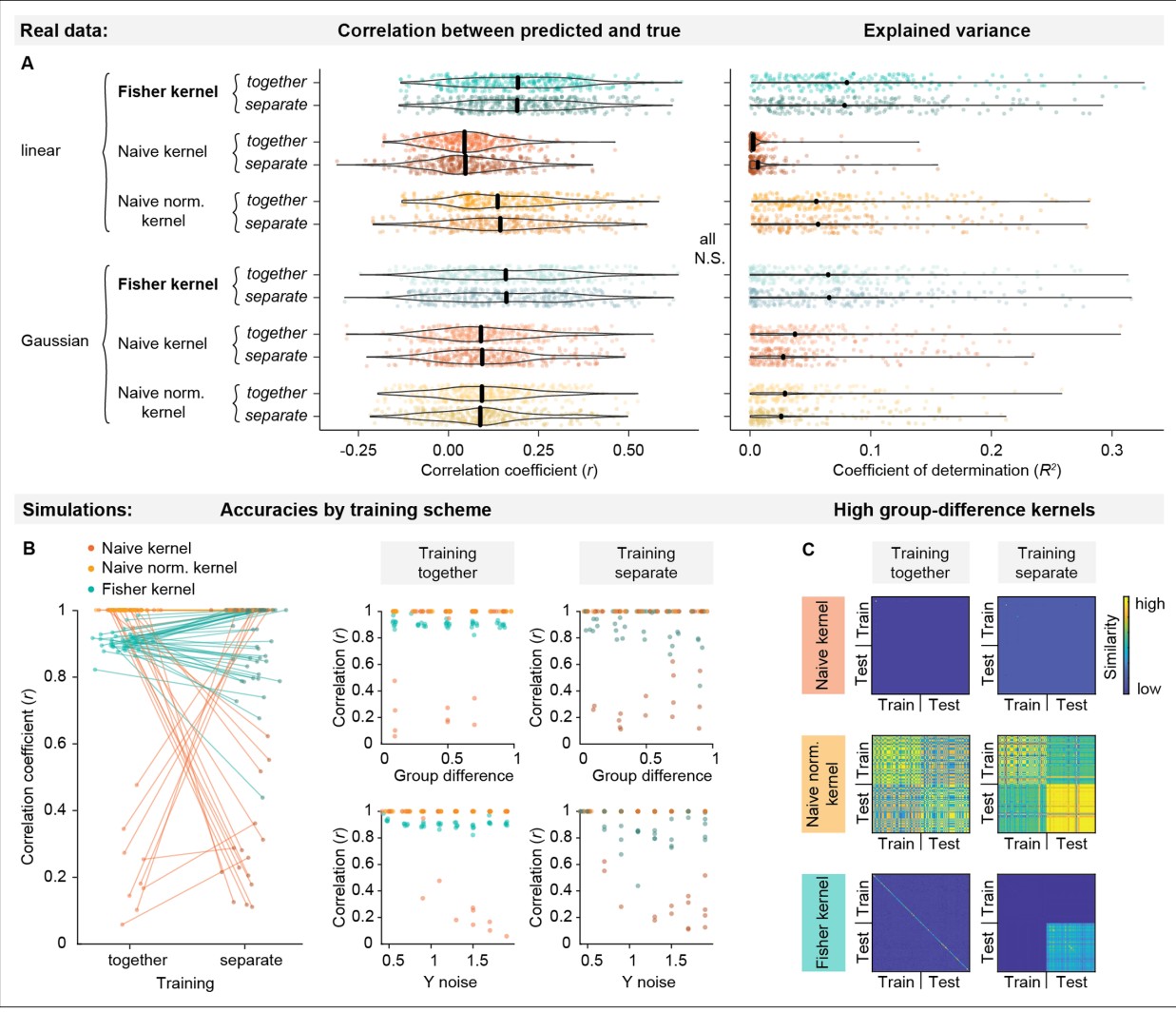

**Figure 6.** Effects of HMM training scheme. (**a**) Prediction accuracies for HMM-based kernels depending on HMM training scheme (training on all subjects: together; training only on training set: separate). In the real data (N=1001), fitting the HMM to all subjects before constructing the kernels compared to fitting it only to the training set to preserve train-test separation has no effect. Note that we are here plotting the fold-wise accuracies (as opposed to averaged over folds, as in the figures above), and we only ran one iteration of CV (rather than 100 repetitions, as in the figures above). (**b**) Prediction accuracies in simulated heterogeneous subject groups depending on training scheme, between-group difference, and target variable (Y) noise. In simulated data, the Fisher kernel's performance decreases when the test subjects are increasingly different from the training subjects. (**c**) Example kernels for high between-group difference. While the naive kernel underperforms in both cases, the strong difference between training and test subjects is visible in the naive normalised kernel, while it completely dominates the Fisher kernel.

since they are all different from the group-level model. This between-group difference may then over-shadow the more subtle differences related to the target variable, as shown in the example kernels in *Figure 6c*. Whether or not this behaviour is desired will depend on the use case.

## Discussion

In this work, we aimed to establish an approach that allows leveraging a rich description of the patterns in which brain activity unfolds over time to predict individual traits. We showed that the HMM-Fisher kernel approach accurately and reliably predicts traits from brain dynamics models trained on neuro-imaging data. It preserves the structure of the underlying brain dynamics model, making it ideal for combining generative and predictive models. We compared the Fisher kernel to kernels which ignore the structure of the brain dynamics model ('naïve' kernels), to a previously used method based on Kullback-Leibler divergence (*Vidaurre et al., 2021*), and to methods based on time-averaged

functional connectivity. The linear Fisher kernel had an overall higher prediction accuracy than all other time-varying methods and several time-averaged methods, though most comparisons were not statistically significant given the narrow margin for improvements. The linear Fisher kernel was also among the most reliable: It never produced excessive errors and was robust to changes in training sets. Like in the time-varying methods, working in Riemannian space also improved prediction for the time-averaged methods, indicating that respecting the geometry of the space that the predictors lie on is an important factor for predictive modelling in neuroscience. While we here focussed on fMRI, the method can also be applied to other modalities like MEG or EEG. It can also be straightforwardly implemented in any kernel-based prediction model or classifier, including kernel ridge regression, support vector machines (SVM), kernel fisher discriminant analysis (k-FDA), kernel logistic regression (KLR), or nonlinear PCA. Indeed, it can also be applied to other probabilistic generative models aside from the HMM, for example Dynamic network modes (*Gohil et al., 2022*).

Our findings were consistent in two alternative settings: when using less data per subject and when modelling only time-varying FC (rather than amplitude and FC states). This supports the generalisability of the results. However, we observed overall lower accuracies when using only one scanning session consisting of 1200 timepoints per subject. This indicates that more available timepoints per subject allow better characterisation of an individual. We also found that the Gaussian versions of the kernels are generally more error-prone and susceptible to changes in the training set, although they may predict more accurately in certain runs. Implementing Gaussian kernels in a predictive model is also computationally more expensive, making them less practical.

While we here tested robustness in terms of susceptibility to changes in CV folds, it remains to be shown to what extent model performance is sensitive to the random initialisation of the HMM, which affects the parameter estimation (*Alonso and Vidaurre, 2023*; *Griffin et al., 2024*). We also showed that the Fisher kernel is most sensitive to changes in state descriptions, that is individual differences in the amplitude or functional connectivity of certain brain states. While this could be a disadvantage if a trait was more closely related to how an individual transitions between brain states, we found that this was not the case in any of the traits we tested here. Other traits than the ones we tested here may of course be more related to individual transition patterns. For this case, we showed in simulations that the Fisher kernel can be modified to recognise changes in transitions if they are of interest for the specific research question.

Finally, we showed that the results we presented here are unaffected by separation between training and test set at the step of training the group-level HMM. However, since the Fisher kernel defines individuals in reference to the group-level model, we showed in simulations that separating training and test subjects prior to fitting the HMM may result in biased kernels that overestimate dissimilarity of the test subjects. This is an important consideration as it may affect studies with small sample sizes and with heterogeneous training and test sets (e.g. where a researcher may want to fit a model to healthy controls and subsequently test it on patients).

We here aimed to show the potential of the HMM-Fisher kernel approach to leverage information from patterns of brain dynamics to predict individual traits in an example fMRI dataset as well as simulated data. The fMRI dataset we used (HCP 1200 Young Adult) is a large sample taken from a healthy, young population, and it remains to be shown how the exhibited performance generalises to other datasets, for example other modalities such as EEG/MEG, clinical data, older populations, different data quality, or smaller sample sizes both in terms of the number of participants and the scanning duration. Additionally, we only tested our approach for the prediction of a specific set of demographic items and cognitive scores; it may be interesting to test the framework also on clinical variables, such as the presence of a disease or the response to pharmacological treatment.

There is growing interest in combining different data types or modalities, such as structural, static, and dynamic measures, to predict phenotypes (*Engemann et al., 2020*; *Schouten et al., 2016*). While directly combining the features from each modality can be problematic, modality-specific kernels, such as the Fisher kernel for time-varying amplitude and/or FC, can be easily combined using approaches such as stacking (*Breiman, 1996*) or Multi Kernel Learning (MKL; *Gönen and Alpaydın, 2011*). MKL can improve prediction accuracy of multimodal studies (*Vaghari et al., 2022*), and stacking has recently been shown to be a useful framework for combining static and time-varying FC predictions (*Griffin et al., 2024*). A detailed comparison of different multimodal prediction strategies including kernels for time-varying amplitude/FC may be the focus of future work.

In a clinical context, while there are nowadays highly accurate biomarkers and prognostics for many diseases, others, such as psychiatric diseases, remain poorly understood, diagnosed, and treated. Here, improving the description of individual variability in brain measures may have potential benefits for a variety of clinical goals, for example to diagnose or predict individual patients' outcomes, find biomarkers, or to deepen our understanding of changes in the brain related to treatment responses like drugs or non-pharmacological therapies (*Marquand et al., 2016*; *Stephan et al., 2017*; *Wen et al., 2022*; *Wolfers et al., 2015*). However, the focus so far has mostly been on static or structural information, leaving the potentially crucial information from brain dynamics untapped. Our proposed approach provides one avenue of addressing this by leveraging individual patterns of time-varying amplitude and FC, as one of many possible descriptions of brain dynamics, and it can be flexibly modified or extended to include, for example information about temporally recurring frequency patterns (*Vidaurre et al., 2016*). In order to be able to use predictive models from brain dynamics in a clinical context, predictions must be reliable, particularly if we want to interpret model errors, as in models of "brain age" (*Cole and Franke, 2017*; *Denissen et al., 2022*; *Smith et al., 2019*). As we demonstrated in this work, there can be extreme errors and large variation in some predictive models, and these issues are not resolved by estimating model performance in a standard cross-validated fashion. We here showed that taking the structure of the underlying model or predictors into account, and thoroughly assessing not only accuracy but also errors and robustness, we can reliably use information from brain dynamics to predict individual traits. This will allow gaining crucial insights into cognition and behaviour from how brain function changes over time, beyond structural and static information.

## Methods
### HCP imaging and behavioural data
We used data from the open-access Human Connectome Project (HCP) S1200 release (*Smith et al., 2013a*; *Van Essen et al., 2013*), which contains MR imaging data and various demographic and behavioural data from 1200 healthy, young adults (age 22–35). All data described below, that is timecourses of the resting-state fMRI data and demographic and behavioural variables, are publicly available at https://db.humanconnectome.org.

Specifically, we used resting state fMRI data of 1001 subjects, for whom any of the behavioural variables of interest were available. Each participant completed four resting state scanning sessions of 14 min and 33 s duration each. This resulted in 4800 timepoints per subject. For the main results, we used all four resting state scanning sessions of each participant to fit the model of brain dynamics (but see *Figure 2—figure supplement 2* for results with just one session). The acquisition parameters are described in the HCP acquisition protocols and in *Van Essen et al., 2013*; *Van Essen et al., 2012*. Briefly, structural and functional MRI data were acquired on a 3T MRI scanner. The resting state fMRI data was acquired using multiband echo planar imaging sequences with an acceleration factor of 8 at 2 mm isotropic spatial resolution and a repetition time (TR) of 0.72 s. The preprocessing and timecourse extraction pipeline is described in detail in *Smith et al., 2013a*. For the resting state fMRI scans, preprocessing consisted of minimal spatial preprocessing and surface projection (*Glasser et al., 2013*), followed by temporal preprocessing. Temporal preprocessing consisted of single-session Independent Component Analysis (ICA; *Beckmann, 2012*) and removal of noise components (*Griffanti et al., 2014*; *Salimi-Khorshidi et al., 2014*). Data were high-pass-filtered with a cut-off at 2000s to remove linear trends.

The parcellation was estimated from the data using multi-session spatial ICA on the temporally concatenated data from all subjects. Note that this means that there is no strict divide between the subjects used for training and the subjects for testing the later predictive models, so that there is potential for leakage of information between training and test set. However, since this step does not concern the target variable, but only the preprocessing of the predictors, the effect can be expected to be minimal (*Rosenblatt et al., 2024*). Using this approach, a data-driven functional parcellation with 50 parcels was estimated, where all voxels are weighted according to their activity in each parcel, resulting in a weighted, overlapping parcellation. While other parcellations are available for the resting-state fMRI HCP dataset, we chose this parcellation because dynamic changes in FC have been shown to be better detected in this parcellation compared to other functional or anatomical parcellations or more fine-grained parcellations (*Ahrends et al., 2022*). Timecourses were extracted

using dual regression (**Beckmann et al., 2009**), where group-level components are regressed onto each subject's fMRI data to obtain subject-specific versions of the parcels and their timecourses. We normalised the timecourses of each subject to ensure that the model of brain dynamics and, crucially, the kernels were not driven by (averaged) amplitude and variance differences between subjects.

Subjects in the HCP study completed a range of demographic and behavioural questionnaires. Following **Vidaurre et al., 2021**, we here focus on a subset of those items, including age and various cognitive variables. The cognitive variables span items assessing memory, executive function, fluid intelligence, language, processing speed, spatial orientation, and attention. The full list of the 35 behavioural variables used here, as well as their categorisation within the HCP dataset can be found in **Supplementary file 1a**.

## The Hidden Markov model

To estimate patterns of time-varying amplitude and FC, we here use the Hidden Markov Model (**Vidaurre et al., 2016**; **Vidaurre et al., 2017**). However, the kernels, which are explained in detail in the following section, can be constructed from any generative probabilistic model.

The Hidden Markov model (HMM) is a generative probabilistic model, which assumes that an observed time-series, such as BOLD signal in a given parcellation, was generated by a sequence of 'hidden states' (**Baum and Eagon, 1967**; **Baum and Petrie, 1966**). We here model the states as multivariate Gaussian distributions, defined both in terms of mean and covariance —which can be interpreted as distinct patterns of amplitude and FC (**Vidaurre et al., 2017**). For comparison, we also considered a second variety of the HMM where state means were pinned to zero (given that the data was demeaned and the global average is zero) and only the covariance was allowed to vary across states (**Vidaurre et al., 2017**), which is shown in **Figure 2—figure supplement 3**. This is equivalent to using a Wishart state model, and therefore focuses more specifically on time-varying FC.

The HMM is described by a set of parameters $\theta$, containing the state probabilities $\pi$, the transition probabilities $A$, the mean vectors $\mu$ of all states (if modelled), and the covariance matrices $\Sigma$ of all states:

$$\theta = [\pi, A, \mu, \Sigma] \qquad \pi \in \mathbb{R}^{1 \times K}, A \in \mathbb{R}^{K \times K}, \mu \in \mathbb{R}^{K \times M}, \Sigma \in \mathbb{R}^{K \times M \times M}$$

where $K$ is the number of states and $M$ is the number of parcels in the parcellation. The entire set of parameters $\theta$ is estimated from the data. The number of states can be understood as the level of detail or granularity with which we describe the spatiotemporal patterns in the data, akin to a dimensionality reduction, where a small number of states will lead to a very general, coarse description and a large number of states will lead to a very detailed, fine-grained description. Here, we chose a small number of states, $K = 6$, to ensure that the group-level HMM states are general enough to be found in all subjects, since a larger number of states increases the chances of certain states being present only in a subset of subjects. The exact number of states is less relevant in this context, since the same HMM estimation is used for all kernels.

The HMM is a probabilistic generative model, as the generative process works by sampling probabilistically in this case from a Gaussian distribution with mean $\mu_k$ and covariance $\Sigma_k$ when state $k$ is active:

$$X_t | q_t = k \sim \mathcal{N}\left(\mu_k, \Sigma_k\right)$$

where $X_t$ is the timeseries at timepoint $t$ and $q_t$ is the currently active state. Which state is active depends on the previous state $q_{t-1}$ and is determined by the transition probabilities $A$, so that the generated state sequence is sampled from a categorical distribution with parameters:

$$q_t | q_{t-1} = k \sim Cat\left(A_k\right)$$

where $A_k$ indicates the k-th row of the transition probability matrix.

The space of parameters $\theta$ forms a Riemannian manifold $R_\theta$, where the relationships between the different parameters of the HMM are acknowledged by construction. The Fisher kernel, as described below, is built upon a projection on this manifold, so predictions based on this kernel account for the mathematical structure of the HMM.

Here, we fit the HMM to the concatenated timeseries of all $N$ subjects (see **Figure 1**, step 1). We refer to the group-level estimate as HMM$^0$, which is defined by the parameters $\theta^0$ (see **Figure 1**, step 2):

$$\theta^0 = \left[\pi^0, A^0, \mu^0, \Sigma^0\right]$$

To use the information from the HMM to predict subjects' phenotypes, we estimate subject-specific versions of the group-level HMM (see **Figure 1**, step 3) through dual estimation (**Vidaurre et al., 2017**). Dual estimation refers to the process of fitting the previously estimated group-level model again to a single subject's timeseries, so that the parameters from the group-level model HMM$^0$ are adapted to fit the individual. We will refer to the subject-specific estimate for subject $n$ as HMM$^n$, with parameters $\theta^n$.

These subject-specific HMM parameters are the features from which we construct the kernels. To understand which features are most important for the predictions, we also construct versions of the kernels that include only subsets of the features. Specifically, we can group the features into two subsets: 1. the state features, describing *what* states look like, containing the mean vectors $\mu$, and the covariance matrices $\Sigma$ of all states, and 2. the transition features, describing *how* individuals transition between these states, containing the initial state probabilities $\pi$ and the transition probabilities $A$. By removing one or the other set of features and evaluating how model performance changes compared to the full kernels, we can draw conclusions about the importance of these two different types of changes for the predictions. Since the state features are considerably more numerous than the transition features (15,300 state features compared to 42 transition features in this case), we also construct a version of the kernels where state features have been reduced to the same number as the transition features using PCA, that is we use all 42 transition features and the first 42 PCs of the state features. This allowed us to perform a fairer comparison of what elements in the model are more predictive of the subject traits.

## Kernels from Hidden Markov models

Kernels (**Shawe-Taylor and Cristianini, 2004**) are a convenient approach to accommodate nonlinearity and to work with high-dimensional, complex features, such as parameters from a model of brain dynamics. In general, kernels are similarity functions, and they can be used straightforwardly in a prediction algorithm. While feature matrices can be very high dimensional, a kernel is represented by a (no. of subjects by no. of subjects) matrix. Kernel methods can readily be adapted to deal with nonlinear decision boundaries in prediction, by projecting the data into a high-dimensional (possibly infinite-dimensional) space through an embedding $x \to \phi_x$; then, by estimating a linear separating hyperplane on this space, we can effectively have a nonlinear estimator on the original space (**Shawe-Taylor and Cristianini, 2004**). In practice, instead of working explicitly in a higher-dimensional embedding space, the so-called kernel trick uses a kernel function $\kappa(n, m)$ containing the similarity between data points $n$ and $m$ (here, subjects) in the higher-dimensional embedding space (**Schölkopf et al., 2002**; **Shawe-Taylor and Cristianini, 2004**), which can be simpler to calculate. Once $\kappa(\cdot, \cdot)$ is computed for each pair of subjects, this is all that is needed for the prediction. This makes kernels computationally very efficient, since in most cases the number of subjects will be smaller than the number of features —which, in the case of HMMs, can be very large (potentially, in the order of millions). However, finding the right kernel can be a challenge because there are many available alternatives for the embedding.

Here, in combination with a linear predictive model, we apply a kernel that is specifically conceived to be used to compare instances of generative models such as the HMM. We expected this to result in better predictions than existing methods. Using the same HMM estimate, we compare three different kernels, which map the HMM parameters into three distinct spaces, corresponding to different embeddings (see **Figure 1**, step 4): the naïve kernel, the naïve normalised kernel, and the Fisher kernel.

While the first two kernels (naïve and naïve normalised kernel) do not take into account constraints imposed by the HMM on how the model parameters can change with respect to each other, the Fisher kernel does. The Fisher kernel achieves this by calculating how, and by how much, one should change the group level HMM parameters to make the model generate data that is more like the data from a particular subject.

We then construct linear and Gaussian versions of the different kernels (see **Figure 1**, step 4), which take the general form $\kappa_l(n,m) = \phi_{x^n}^T \phi_{x^m}$ for the linear kernel and $\kappa_g(n,m) = \exp(-\frac{\|\phi_{x^n} - \phi_{x^m}\|^2}{2\tau^2})$ for the Gaussian kernel. We compare these to a kernel constructed using Kullback-Leibler divergence, previously used for predicting behavioural phenotypes (**Vidaurre et al., 2021**).

## Naïve kernel

The naïve kernel is based on a simple vectorisation of the subject-specific version of the HMM's parameters, each on their own scale. This means that the kernel does not take relationships between the parameters into account and the parameters are here on different scales. This procedure can be thought of as computing Euclidean distances between two sets of HMM parameters, ignoring the actual geometry of the space of parameters. For each subject $n$, we vectorise parameters $\theta^n$ obtained through dual estimation of the group-level parameters $\theta^0$ to map the example $x^n \rightarrow \phi_{x^n}$ to:

$$\theta^n = \left(\pi^n, A^n, \mu^n, \Sigma^n\right) \qquad \theta^n \in \mathbb{R}^{1 \times \left(K + K*K + K*M + K*M*M\right)}$$

We will refer to this vectorised version of the subject-specific HMM parameters as "naïve $\theta$". The naïve $\theta$ are the features used in the naïve kernel $\kappa_N$. We first construct a linear kernel from the naïve $\theta$ features using the inner product of the feature vectors. The linear naïve kernel $\kappa_{Nl}$ between subjects $n$ and $m$ is thus defined as:

$$\kappa_{Nl}(n,\, m) = \left\langle \theta^n, \theta^m \right\rangle \qquad \kappa_{Nl} \in \mathbb{R}^{N \times N}$$

where $\langle \theta^n, \theta^m \rangle$ denotes the inner product between $\theta^n$ and $\theta^m$. Using the same feature vectors, we can also construct a Gaussian kernel from the naïve $\theta$. The Gaussian naïve kernel $\kappa_{Ng}$ for subjects $n$ and $m$ is defined as:

$$\kappa_{Ng}(n,\, m) = \exp\left(-\frac{\|\theta^n - \theta^m\|^2}{2\tau^2}\right) \qquad \kappa_{Ng} \in \mathbb{R}^{N \times N}$$

where $\tau$ is the radius of the radial basis function, and $\|\theta^n - \theta^m\|$ is the $L_2$-norm of the difference of the feature vectors (naïve $\theta$ for subjects $n$ and $m$). Compared to a linear kernel, a Gaussian kernel embeds the features into a more complex space, which can potentially improve the accuracy. However, this kernel has an additional parameter $\tau$ that needs to be chosen, typically through cross-validation. This makes a Gaussian kernel computationally more expensive and, if the additional parameter $\tau$ is poorly estimated, more error-prone. The effect of the hyperparameters on errors is shown in **Figure 2— figure supplement 4**.

While the naïve kernel takes all the information from the HMM into account by using all parameters from a subject-specific version of the model, it uses these parameters in a way that ignores the structure of the model that these parameters come from. In this way, the different parameters in the feature vector are difficult to compare, since for example a change of 0.1 in the transition probabilities between two states is not of the same magnitude as a change of 0.1 in one entry of the covariance matrix of a specific state. In the naïve kernel, these two very different types of changes would be treated indistinctly.

## Naïve normalised kernel

To address the problem of parameters being on different scales, the naïve normalised kernel makes the scale of the subject-specific vectorised parameters (i.e. the naïve $\theta$) comparable across parameters. Here, the mapping $x \rightarrow \phi_x$ consists of a vectorisation and normalisation across subjects of the subject-specific HMM parameters, by subtracting the mean over subjects from each parameter and dividing by the standard deviation. This kernel does not respect the geometry of the space of parameters either.

As for the naïve kernel, we can then construct a linear kernel from these vectorised, normalised parameters $\widetilde{\theta}$ by computing the inner product for all pairs of subjects $n$ and $m$ to obtain the linear naïve normalised kernel $\kappa_{NNl}$:

$$\kappa_{NNl}(n,\, m) = \left\langle \tilde{\theta}^n, \tilde{\theta}^m \right\rangle \qquad \kappa_{NNl} \in \mathbb{R}^{N \times N}$$

We can also compute a Gaussian kernel from the naïve normalised feature vectors to obtain the Gaussian version of the naïve normalised kernel $\kappa_{NNg}$:

$$\kappa_{NNg}(n, m) = \exp\left(-\frac{\left\|\tilde{\theta}^n - \tilde{\theta}^m\right\|^2}{2\tau^2}\right) \qquad \kappa_{NNg} \in \mathbb{R}^{N\mathrm{x}N}$$

In this way, we have constructed a kernel in which parameters are all on the same scale, but which still ignores the complex relationships between parameters originally encoded by the underlying model of brain dynamics.

## Fisher kernel

The Fisher kernel (*Jaakkola et al., 1999*; *Jaakkola and Haussler, 1998*) is specifically designed to preserve the structure of a generative probabilistic model (here, the HMM). This can be thought of as a 'proper' projection on the manifold, as illustrated in *Figure 1*, step 4b. Similarity between subjects is here defined in reference to a group-level model of brain dynamics. The mapping $x \rightarrow \phi_x$ is given by the 'Fisher score', which indicates how (i.e., in which direction in the Riemannian parameter space) we would have to change the group-level model to better explain a particular subject's timeseries. The similarity between subjects can then be described based on this score, so that two subjects are defined as similar if the group-level model would have to be changed in a similar direction for both, and dissimilar otherwise.

More precisely, the Fisher score is given by the gradient of the log-likelihood with respect to each model parameter:

$$g(\theta^0, x^n) = \left(\frac{\partial \log \mathcal{L}_{\theta^0}(x^n)}{\partial \theta^0}\right) \qquad g \in \mathbb{R}^{1\mathrm{x}\left(K+K*K+K*M+K*M*M\right)}$$

where $x^n$ is the timeseries of subject $n$, and $\mathcal{L}_\theta(x) = P(x|\theta)$ represents the likelihood of the timeseries $x$ given the model parameters $\theta$. This way, the Fisher score maps an example (i.e. a subject's timeseries) $x^n$ into a point in the gradient space of the Riemannian manifold $R_\theta$ defined by the HMM parameters.

The invariant Fisher kernel $\kappa_{F-}$ is the inner product of the Fisher score $g$, scaled by the Fisher information matrix $F$, which gives a local metric on the Riemannian manifold $R_\theta$:

$$\kappa_F - (n, m) = g(\theta^0, x^n)^T F_{R_\theta}^{-1} g(\theta^0, x^m) \qquad \kappa_{F-} \in \mathbb{R}^{N\mathrm{x}N}$$

for subjects $n$ and $m$. $F_{R_\theta}$ is the Fisher information matrix, defined as

$$F_{R_\theta} = \mathbb{E}_x \left[g\left(\theta^0, x\right) g\left(\theta^0, x\right)^{\mathrm{T}}\right]$$

where the expectation is with respect to $x$ under the distribution $P(x|\theta)$. The Fisher information matrix $F_{R_\theta}$ can be approximated empirically:

$$\hat{F}_{R_\theta} = \frac{1}{N} \sum_{i=1}^{N} g\left(\theta^0, x^i\right) g\left(\theta^0, x^i\right)^{\mathrm{T}}$$

which is simply the covariance matrix of the gradients $g$. Using $F_{R_\theta}$ essentially serves to whiten the gradients; therefore, given the large computational cost associated with $F_{R_\theta}$, we here disregard the Fisher information matrix and reduce the invariant Fisher kernel to the so-called practical Fisher kernel (*Jaakkola et al., 1999*; *Jaakkola and Haussler, 1998*; *van der Maaten, 2011*), for which the linear version $\kappa_{Fl}$ takes the form:

$$\kappa_{Fl}(n, m) = \left\langle g(\theta^0, x^n), g(\theta^0, x^m)\right\rangle \qquad \kappa_{Fl} \in \mathbb{R}^{N\mathrm{x}N}$$

In this study, we will use the practical Fisher kernel for all computations.

One issue when working with the linear Fisher kernel is that the gradients of typical examples (i.e. subjects whose timeseries can be described by similar parameters as the group-level model) are close to zero, while gradients of atypical examples (i.e. subjects who are very different from the group-level

model) can be very large. This may lead to an underestimation of the similarity between two typical examples because their inner product is very small even though they are very similar. To mitigate this, we can plug the gradient features (i.e. the Fisher scores $g$) into a Gaussian kernel, which essentially normalises the kernel. For subjects $n$ and $m$, where $x^n$ is the timeseries of subject $n$ and $x^m$ is the timeseries of subject $m$, the Gaussian Fisher kernel $\kappa_{Fg}$ is defined as

$$\kappa_{Fg}(n,m) = \exp\left(-\frac{\left\|g(\theta^0, x^n) - g(\theta^0, x^m)\right\|^2}{2\tau^2}\right) \qquad \kappa_{Fg} \in \mathbb{R}^{N \times N}$$

where $\|g(\theta^0, x^n) - g(\theta^0, x^m)\|$ is the distance between examples $n$ and $m$ in the gradient space, and $\tau$ is the width of the Gaussian kernel.

## Kullback-Leibler divergence

The Kullback-Leibler (KL) divergence is an information-theoretic distance measure which estimates divergence between probability distributions —in this case between subject-specific versions of the HMM. Here, KL divergence of subject $n$ from subject $m$, $KL(\text{HMM}^n \| \text{HMM}^m)$ can be interpreted as how much new information the HMM of subject $n$ contains if the true distribution was the HMM of subject $m$. KL divergence is not symmetric, that is $KL(\text{HMM}^n \| \text{HMM}^m)$ is different than $KL\left(\text{HMM}^n \| \text{HMM}^m\right)$. We here use an approximation of KL divergence as in *Do, 2003* and *Vidaurre et al., 2021*. That is, given two models $\text{HMM}^n$ from $\text{HMM}^m$ for subject $n$ and subject $m$, we have

$$KL\left(\text{HMM}^n \| \text{HMM}^m\right) = \sum_k v_k KL\left(A_k^n, A_k^m\right) + v_k KL\left(G_k^n, G_k^m\right)$$

where $A_k^n$ are the transition probabilities from state $k$ into any other state according to $\text{HMM}^n$ and $G_k^n$ are the state Gaussian distributions for state $k$ and $\text{HMM}^n$ (respectively $A_k^m$ and $G_k^m$ for $\text{HMM}^m$); see *MacKay et al., 2003*. Since the transition probabilities are Dirichlet-distributed and the state distributions are Gaussian distributed, KL divergence for those has a closed-form solution. Variables $v_k$ can be computed numerically such that

$$vA^n = v,$$
$$\lim_{x \to \infty} \pi^n A^{nx} = \nu$$

To be able to use KL divergence as a kernel, we symmetrise the KL divergence matrix as

$$\mathcal{D}_{KL}(n,m) = 0.5 \, KL(\text{HMM}^n \| \text{HMM}^m) + 0.5 \, KL(\text{HMM}^n \| \text{HMM}^m) \qquad \mathcal{D}_{KL} \in \mathbb{R}^{N \times N}$$

This symmetrised KL divergence can be plugged into a radial basis function, analogous to the Gaussian kernels to obtain a similarity matrix $\kappa_{KL}$

$$\kappa_{KL}(n,m) = \exp(-\frac{(\mathcal{D}_{KL}(n,m))^2}{2\tau^2}) \qquad \kappa_{KL} \in \mathbb{R}^{N \times N}$$

The resulting KL similarity matrix can be used in the predictive model in a similar way as the kernels described above.

## Predictive model: Kernel ridge regression

Similarly to *Vidaurre et al., 2021*, we use kernel ridge regression (KRR) to predict demographic and behavioural variables from the different kernels (other kernel-based prediction models or classifiers such as a support vector machine are also possible). KRR is the kernelised version of ridge regression (*Saunders et al., 1998*):

$$\hat{y} = h\alpha \qquad \alpha \in \mathbb{R}^{S_{train} \times 1}, \; h \in \mathbb{R}^{S_{test} \times S_{train}}$$

where $\alpha$ are the regression weights; $h$ is the (number of subjects in test set by number of subjects in training set) kernel matrix between the subjects in the training set and the subjects in the test set; $y$ are the predictions in the (out-of-sample) test set; $S_{train}$ are the number of subjects in the training set;

and $S_{test}$ are the number of subjects in the test set. The regression weights α can be estimated using the kernels specified above as

$$\alpha = (\kappa + \lambda I)^{-1} * y \qquad\qquad \kappa \in \mathbb{R}^{S_{train} \times S_{train}}$$

where $\lambda$ is a regularisation parameter that we can choose through cross-validation; $I$ is the identity matrix; $\kappa$ is the ($S_{train}$ by $S_{train}$) kernel matrix of the subjects in the training set; and $y$ are the training examples.

We use KRR to separately predict each of the 35 demographic and behavioural variables from each of the different methods, removing subjects with missing entries from the prediction. We used k-fold nested cross-validation (CV) to select and evaluate the models. We used 10 folds for both the outer loop (used to train and test the model) and the inner loop (used to select the optimal hyperparameters) such that 90% were used for training and 10% for testing. The optimal hyperparameters $\lambda$ (and

$\tau$ in the case of the Gaussian kernels) were selected using grid-search from the vectors $\lambda$ =[0.0001, 0.001, 0.01, 0.1, 0.3, 0.5, 0.7, 0.9, 1] and $\tau$ =[1/5, 1/3, 1/2, 1, 2, 3, 5]. In both the outer and the inner loop, we accounted for family structure in the HCP dataset so that subjects from the same family were never split across folds (**Winkler et al., 2015**). Within the CV, we regressed out sex and head motion confounds, that is we estimated the regression coefficients for the confounds on the training set and applied them to the test set (**Snoek et al., 2019**). We repeated the nested 10-fold CV 100 times, so that different combinations of subjects were randomly assigned to the folds at each new CV iteration to obtain a distribution of model performance values for each variable. This is to explicitly show how susceptible each model was to changes in the training folds, which we can take as a measure of the robustness of the estimators, as described below. We generated the 100 random repetitions of the 10 outer CV folds once, and then used them for training and prediction of all methods, so that all methods were fit to the same partitions.

## Evaluation criteria

We evaluate the models in terms of two outcome criteria: prediction accuracy and reliability. For prediction accuracy, we used Pearson's correlation coefficient $r(\hat{y}, y)$ between the model-predicted values $y$ and the actual values $y$ of each variable and the coefficient of determination $R^2$. The second criterion, reliability, concerns two aspects: (i) that the model will never show excessively large errors for single subjects that could harm interpretation; and (ii) that the model's accuracy will be consistent across random variations of the training set —in this case by using different (random) iterations for the CV folds. This is important if we want to interpret prediction errors for example in clinical contexts, which assumes that the error size of a model in a specific subject reflects something biologically meaningful, for example whether a certain disease causes the brain to 'look' older to a model than the actual age of the subject (**Denissen et al., 2022**). Maximum errors inform us about single cases where a model (that may typically perform well) fails. The maximum absolute error (MAXAE) is the single largest error made by the regression model in each iteration, that is

$$MAXAE = \max_{i \in N} \left( \left| y_i - \hat{y}_i \right| \right)$$

Since the traits we predict are on different scales, the MAXAE is difficult to interpret, for example a MAXAE of 10 would be considered small if the true range of the variable we are predicting was 1,000, while it would be considered large if the true range of the variable was 1. To make the results comparable across the different traits, we therefore normalise the MAXAE by dividing it by the range of the respective variable. In this way, we obtain the NMAXAE:

$$NMAXAE = \frac{MAXAE}{y_{max} - y_{min}}$$

Since the NMAXAEs follow extreme value distributions, it is more meaningful to consider the proportion of the values exceeding relevant thresholds than testing for differences in the means of these distributions (**Gumbel, 1958**). We here consider the risk of large errors (NMAXAE >10), very large errors (NMAXAE >100), and extreme errors (NMAXAE >1000) as the percentage of runs (across variables and CV iterations) where the model's NMAXAE exceeds the given threshold. Since NMAXAE

is normalised by the range of the actual variable, these thresholds correspond to one, two, and three orders of magnitude of the actual variable's range. If we are predicting age, for instance, and the true ages of the subjects range from 25 years to 35 years, an NMAXAE of 1 would mean that the model's least accurate prediction is off by 10 years, an NMAXAE of 10 would mean that the least accurate prediction is off by 100 years, an NMAXAE of 100 would be off by 1000 years, and an NMAXAE of 1,000 would be off by 10,000 years. A model that makes such large errors, even in single cases, would be unusable for interpretation. Our reliability criterion in terms of maximum errors is therefore that the risk of large errors (NMAXAE >10) should be 0%.

For a model to be reliable, it should also be robust in the sense of susceptibility to changes in the training examples. Robustness is an important consideration in prediction studies, as it determines how reproducible a study or method is — which is often a shortcoming in neuroimaging-based prediction studies (*Varoquaux et al., 2017*). We evaluated the robustness by iterating nested 10-fold CV 100 times for each variable, randomising the subjects in the folds at each iteration, so that the models would encounter a different combination of subjects in the training and test sets each time they are run. Looking at this range of 1000 accuracies (10 folds * 100 repetitions) for each variable, we can assess whether the model's performance changes drastically depending on which combinations of subjects it encountered in the training phase, or whether the performance is the same regardless of the subjects encountered in the training phase. The former would be an example of a model that is susceptible to changes in training examples and the latter an example of a model that is robust. We here quantify robustness as the standard deviation (S.D.) of the prediction accuracy $r$ across the 10 folds and 100 CV iterations of each variable, where a small mean S.D. over variables indicates higher robustness.

We test for significant differences in mean prediction accuracy between the methods using repeated k-fold cross-validation corrected t-tests (*Bouckaert and Frank, 2004*). For results obtained through cross-validation with $k$ folds and $r$ repetitions, the corrected t-statistic is calculated as:

$$t_{rkCV} = \frac{\frac{1}{k \cdot r} \sum_{i=1}^{k} \sum_{j=1}^{r} x_{ij}}{\sqrt{\left( \frac{1}{k \cdot r} + \frac{n_2}{n_1} \right) \sigma^2}}$$

where $x$ is the difference in prediction accuracy, $n_1$ is the number of samples in the training data, $n_2$ the number of samples in the test data, and $\sigma^2$ the variance estimate. To test for significant differences in mean robustness (S.D. across CV repetitions), we use paired t-tests. All p-values are corrected across multiple comparisons using Benjamini-Hochberg correction to control the false discovery rate (FDR). For the main results, we separately compare the linear Fisher kernel to the other linear kernels, and the Gaussian Fisher kernel to the other Gaussian kernels, as well as to each other. We also compare the linear Fisher kernel to all time-averaged methods. Finally, to test for the effect of tangent space projection for the time-averaged FC prediction, we also compare the Ridge regression model to the Ridge Regression in Riemannian space. To test for effects of removing sets of features, we use the approach described above to compare the kernels constructed from the full feature sets to their versions where features were removed or reduced. Finally, to test for effects of training the HMM either on all subjects or only on the subjects that were later used as training set, we compare each kernel to the corresponding kernel constructed from HMM parameters, where training and test set were kept separate.

## Models based on time-averaged FC features

To compare our approach's performance to simpler methods that do not take dynamics into account, we compared them to seven different regression models based on time-averaged FC features. For each subject, time-averaged FC was computed as covariance between each pair of regions. The first time-averaged model is, analogous to the HMM-derived KL divergence model, a time-averaged KL divergence model. The model is described in detail in *Vidaurre et al., 2021*. Briefly, we construct symmetrised KL divergence matrices of each subject's time-averaged FC and predict from these matrices using the KRR pipeline described above. We refer to this model as time-averaged KL divergence. We next used a kernel constructed from a geodesic distance (i.e. a metric defined on the

Riemannian manifold) of time-averaged covariance matrices. Specifically, we used a Gaussian kernel on the log-Euclidean distance, which is the Frobenius norm of the logarithm map of the time-averaged covariance matrices (*Jayasumana et al., 2013*). We refer to this model as log-Euclidean. The other five time-averaged FC benchmark models do not involve kernels but predict directly from the features instead. Namely, we use two variants of a Ridge Regression and an Elastic Net model (*Zou and Hastie, 2005*), one using the unwrapped time-averaged FC matrices as input (i.e. in Euclidean space), and one using the time-averaged FC matrices in Riemannian space, where covariance matrices are projected into tangent space (*Barachant et al., 2013*; *Smith et al., 2013b*), which we refer to as Ridge Reg. and Ridge Reg. (Riem), and Elastic Net and Elastic Net (Riem.), respectively. Finally, we compare our models to the approach taken in *Rosenberg et al., 2016*, where relevant edges of the time-averaged FC matrices are first selected, and then used as predictors in a regression model. We refer to this model as Selected Edges. All time-averaged FC models are fitted using the same (nested) cross-validation strategy as described above (10-fold CV using the outer loop for model evaluation and the inner loop for model selection using grid-search for hyperparameter tuning, accounting for family structure in the dataset, and repeated 100 times with randomised folds).

## Simulations

### Feature importance

To further understand the behaviour of the different kernels, we simulate data and compare the kernels' ability to recover the ground truth. Specifically, we aim to understand which type of parameter change the kernels are most sensitive to. We generate timeseries for two groups of subjects, timeseries $X^1$ for group 1 and timeseries $X^2$ for group 2, from two separate HMMs with respective sets of parameters $\theta^1$ and $\theta^2$:

$$X^1 \sim \text{HMM}\left(\theta^1\right)$$

$$X^2 \sim \text{HMM}(\theta^2) \qquad X^1, X^2 \in \mathbb{R}^{Mx\left(\left(\frac{N}{2}\right)*T\right)}$$

We simulate timeseries from HMMs with these parameters through the generative process described in 4.2.

For the simulations, we use the group-level HMM of the real dataset with $K = 6$ states used in the main text as basis for group 1, that is $\text{HMM}\left(\theta^1\right) = \text{HMM}\left(\theta^0\right)$. We then manipulate two different types of parameters, the state means $\mu$ and the transition probabilities $A$, while keeping all remaining parameters the same between the groups. In the first case, we manipulate one state's mean between the groups, that is:

$$\theta^1 = \left[\pi^1, A^1, \mu^1, \Sigma^1\right]$$

$$\theta^2 = \left[\pi^1, A^1, \mu^2, \Sigma^1\right]$$

where $\mu^2$ is obtained by simply adding a Gaussian noise vector to the state mean vector of one state:

$$\mu_1^2 = \mu_1^1 + \varphi$$

Here, $\varphi$ is the Gaussian noise vector of size 1 x $M$, $M$ is the number of parcels, here 50, and $\mu_1^1$ and $\mu_1^2$ are the first rows (corresponding to the first state) of the state mean matrices for groups 1 and 2, respectively. We control the amplitude of $\varphi$ so that the difference between $\mu_1^2$ and $\mu_1^1$ is smaller than the minimum distance between any pair of states within one HMM. This is to ensure that the HMM recovers the difference between groups as difference in one state's mean vector, rather than detecting a new state for group 2 that does not occur in group 1 and consequently collapsing two other states. Since the state means $\mu$ and the state covariances $\Sigma$ are the first- and second-order parameters of the same respective distributions, it is to be expected that, although we only directly manipulate $\mu$, $\Sigma$ changes as well.

In the second case, we manipulate the transition probabilities for one state between the groups, while keeping all other parameters the same, i.e.:

$$\theta^1 = \left[\pi^1, A^1, \mu^1, \Sigma^1\right]$$

$$\theta^2 = \left[\pi^1, A^2, \mu^1, \Sigma^1\right]$$

where $A^2$ is obtained by randomly permuting the probabilities of one state to transition into any of the other states, excluding self-transition probability:

$$A^2_{1,1} = A^1_{1,1}$$

$$A^2_{1,2:\ K} = pA^1_{1,2:\ K}$$

Here, $p$ is a random permutation vector of size $(K-1)$ x 1, $A^1_{1,1}$ and $A^2_{1,1}$ are the self-transition probabilities of state 1, and $A^1_{1,2:\ K}$ and $A^2_{1,2:\ K}$ are the probabilities of state 1 to transition into each of the other states.

We then concatenate the generated timeseries

$$X = [X^1, X^2] \qquad\qquad X \in \mathbb{R}^{Mx\,(N*T)}$$

containing 100 subjects for group 1 and 100 subjects for group 2, with 1200 timepoints and 50 parcels per subject. Note that we do not introduce any differences between subjects within a group, so that the between-group difference in HMM parameters should be the most dominant distinction and easily recoverable by the classifiers. We then apply the pipelines described above, running a new group-level HMM and constructing the linear versions of the naïve kernel, the naïve normalised kernel, and the Fisher kernel on these synthetic time series. The second case of simulations, manipulating the transition probabilities, only introduces a difference in few $(K - 1)$ features and keeps the majority of the features the same between the groups, while the first case introduces a difference in a large number of features ($M$ features directly, by changing one state's mean vector, and an additional $M$ x $M$ features indirectly, as this state's covariance matrix will also be affected). To account for this difference, we additionally construct a version of the kernels for the second case of simulations that includes only $\pi$ and $A$, removing the state parameters $\mu$ and $\Sigma$.

Finally, we use a support vector machine (SVM) in combination with the different kernels to recover the group labels and measure the error produced by each kernel. We repeat the whole process (generating timeseries, constructing kernels, and running the SVM) 10 times, in each iteration randomising on three levels: generating new random noise/permutation vectors to simulate the timeseries, randomly initialising the HMM parameters when fitting the group-level HMM to the simulated timeseries, and randomly assigning subjects to 10 CV folds in the SVM.

## Separating training and test set

We also simulate data to understand the sensitivity of the different kernels to separating training and test set before or after running the HMM. While separating training and test set is generally considered to be the gold standard to avoid leakage of information, this strategy may cause issues when the training and test set are very different from each other. In the simplest case, this would cause models to poorly generalise, which we aim to assess with the train-test split. However, in the case of the Fisher kernel, features are not independent but defined in reference to a group set of parameters. That means that, if we train the HMM only on the training set and then construct the Fisher kernel of a group of test subjects that are very different from the training set, this difference will be overrepresented in the kernel and likely overshadow other, more subtle differences between individuals.

Similar to the feature importance simulations, we generate time courses for two groups of subjects from two HMMs: timeseries $X^{train}$ for the training set and timeseries $X^{test}$ for the test set, from HMMs with parameters $\theta^{train}$ and $\theta^{test}$:

$$X^{train} \sim \mathrm{HMM}\left(\theta^{train}\right)$$

$$X^{test} \sim \mathrm{HMM}(\theta^{test}) \qquad\qquad X^{train}, X^{test} \in \mathbb{R}^{Mx\left(\left(\frac{N}{2}\right)*T\right)}$$

We use the group-level HMM fit to real data as the basis for the simulations, so that $\text{HMM}\left(\theta^{\text{train}}\right) = \text{HMM}\left(\theta^{0}\right)$. For the test set, we then add different levels of noise to one of the state's mean vector to simulate varying degrees of between-group difference:

$$\theta^{train} = \left[\pi^{train}, A^{train}, \mu^{train}, \Sigma^{train}\right]$$

$$\theta^{test} = \left[\pi^{train}, A^{train}, \mu^{test}, \Sigma^{train}\right]$$

$$\mu^{test} = \begin{bmatrix} \mu_1^{train} + b\epsilon \\ \mu_2^{train} \\ ... \\ \mu_k^{train} \end{bmatrix}$$

where $\epsilon$ is the 1 x $M$ Gaussian noise vector representing the heterogeneity between the training and the test set and $b$ is a scalar controlling the noise level.

We randomly generate a continuous target variable, $Y$, which the models will aim to predict. We will simulate a single state's mean to be correlated with this target variable, but we will add varying degrees of noise, which will make it gradually more difficult to predict the target. We do this by adding a fraction of the target variable and the noise to a second state mean vector for each subject in the training and the test set:

$$\theta^{s} = \left[\pi^{s}, A^{s}, \mu^{s}, \Sigma^{s}\right]$$

$$\mu^{s} = \begin{bmatrix} \mu_1 \\ \mu_2 + \dfrac{Y^s}{10} + c\varphi^s \\ ... \\ \mu_k \end{bmatrix}$$

where $\mu_k$ is the 1 x $M$ state mean vector for state $k$, $Y^s$ is the simulated target variable and $\varphi^s$ is the 1 x $M$ Gaussian noise vector, controlled by the scaler $c$. That means that the two effects (the target variable and the difference between training and test set) are represented in two separate states' parameters. The models should be able to retrieve the variable of interest in one state while ignoring the between-group noise affecting another state.

We then simulate individual subjects' timeseries for 50 training subjects and 50 test subjects, varying the values for $b$ (controlling the level of between-group difference) in the range $b = \left[0.1, 0.3, 0.5, 0.7, 0.9\right]$ and $c$ (controlling the level of noise on the target variable $Y$) in the range $c = \left[0.5, 0.7, 0.9, 1.1, 1.3, 1.5, 1.7, 1.9\right]$. We then use two different training schemes: the first one where we train the HMM on all subjects (training and test set) before constructing the features (training: together) and the second one where we train the HMM only on the training subjects (training: separate). For the Fisher kernel, that means that the features for the subjects in the test set will be computed in reference to a group-level HMM which they are either part of (training: together) or not a part of (training: separate). We then use the same pipeline described above to predict the simulated target variables from the linear naïve kernel, the linear naïve normalised kernel, and the linear Fisher kernel.

## Implementation

All code used in this paper, including scripts to reproduce the figures and additional application examples of the Fisher Kernel can be found in the repository https://github.com/ahrends/FisherKernel (copy archived at **Ahrends, 2025**). The HMM-Fisher kernel pipeline in Matlab is also publicly available within the repository of the HMM-MAR toolbox at https://github.com/OHBA-analysis/HMM-MAR, (copy archived at **OHBA Analysis Group, 2025**). A Python-version of the Fisher kernel is also available at https://github.com/vidaurre/glhmm (copy archived at **Vidaurre, 2025a**). Code for Ridge Regression and Elastic Net prediction from time-averaged FC features is available at https://github.com/vidaurre/NetsPredict/blob/master/nets_predict5.m (copy archived at **Vidaurre, 2025b**). The procedure for Selected Edges time-averaged FC prediction is described in detail in **Shen et al., 2017** and code is provided at https://www.nitrc.org/projects/bioimagesuite/behavioralprediction.m.

## Acknowledgements

Data were provided by the Human Connectome Project, WU-Minn Consortium (Principal Investigators: David Van Essen and Kamil Ugurbil; 1U54MH091657) funded by the 16 NIH Institutes and Centers that support the NIH Blueprint for Neuroscience Research; and by the McDonnell Center for Systems Neuroscience at Washington University. DV is supported by a Novo Nordisk Foundation Emerging Investigator Fellowship (NNF19OC-0054895) and an ERC Starting Grant (ERC-StG-2019–850404). This research was funded in part by the Wellcome Trust (215573/Z/19/Z). For the purpose of Open Access, the author has applied a CC BY public copyright licence to any Author Accepted Manuscript version arising from this submission. We thank Ben Griffin and Steve Smith for useful discussions and technical collaboration.

## Additional information

### Funding

| Funder | Grant reference number | Author |
| --- | --- | --- |
| Novo Nordisk Fonden | NNF19OC-0054895 | Diego Vidaurre |
| European Research Council | ERC-StG-2019-850404 | Diego Vidaurre |
| Wellcome Trust | 215573/Z/19/Z | Mark W Woolrich |

The funders had no role in study design, data collection and interpretation, or the decision to submit the work for publication. For the purpose of Open Access, the authors have applied a CC BY public copyright license to any Author Accepted Manuscript version arising from this submission.

### Author contributions

Christine Ahrends, Conceptualization, Software, Formal analysis, Investigation, Visualization, Methodology, Writing – original draft; Mark W Woolrich, Methodology, Writing – review and editing; Diego Vidaurre, Conceptualization, Software, Supervision, Funding acquisition, Investigation, Methodology, Writing – review and editing

### Author ORCIDs

Christine Ahrends ⓘ https://orcid.org/0000-0002-9287-1254
Diego Vidaurre ⓘ https://orcid.org/0000-0002-9650-2229

Reviewer #1 (Public review): https://doi.org/10.7554/eLife.95125.3.sa1
Reviewer #3 (Public review): https://doi.org/10.7554/eLife.95125.3.sa2
Author response https://doi.org/10.7554/eLife.95125.3.sa3

## Additional files

### Supplementary files

Supplementary file 1. Overview of target variables (a) and summary of model performance (b).
MDAR checklist

### Data availability

Data were provided by the Human Connectome Project, WU-Minn Consortium (Principal Investigators: David Van Essen and Kamil Ugurbil; 1U54MH091657) funded by the 16 NIH Institutes and Centers that support the NIH Blueprint for Neuroscience Research; and by the McDonnell Center for Systems Neuroscience at Washington University.

The following previously published dataset was used:

| Author(s) | Year | Dataset title | Dataset URL | Database and Identifier |
|---|---|---|---|---|
| Van Essen DC, Smith SM, Barch DM, Behrens TE, Yacoub E, Ugurbil K | 2013 | The WU-Minn Human Connectome Project | https://db.humanconnectome.org | humanconnectome, humanconnectome |

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
