## [Editor Report · eLife Assessment]

This **important** study combines the use of Fisher Kernels with Hidden Markov models aiming to improve brain-behaviour prediction. The evidence supporting the authors' conclusions is **compelling**, comparing brain-behaviour prediction accuracies across a range of different traits, including out of sample assessment. This work is timely and will be of interest to neuroscientists working on functional connectivity for brain-behaviour association.

---

## [Referee Report · Reviewer #1 (Public review)]

Summary:

The authors attempt to validate Fisher Kernels on the top of HMM as a way to better describe human brain dynamics at resting-state. The objective criterion was the better prediction of the proposed pipeline of the individual traits.

Comments on revisions:

The authors addressed adequately all my comments.

---

## [Referee Report · Reviewer #3 (Public review)]

Summary:

In this work, the authors use a Hidden Markov Model (HMM) to describe dynamic connectivity and amplitude patterns in fMRI data, and propose to integrate these features with the Fisher kernel to improve the prediction of individual traits. The approach is tested using a large sample of healthy young adults from the Human Connectome Project. The HMM-Fisher Kernel approach was shown to achieve higher prediction accuracy with lower variance on many individual traits compared to alternate kernels and measures of static connectivity. As an additional finding, the authors demonstrate that parameters of the HMM state matrix may be more informative in predicting behavioral/cognitive variables in this data compared to state-transition probabilities.

Comments on revisions:

The authors have now addressed my comments, and I believe this work will be an interesting contribution to the literature.

---

## [Author Response]

The following is the authors’ response to the original reviews.

**Public Reviews:**

**Reviewer #1 (Public Review):**
Summary:The authors attempt to validate Fisher Kernels on the top of HMM as a way to better describe human brain dynamics at resting state. The objective criterion was the better prediction of the proposed pipeline of the individual traits.Strengths:The authors analyzed rs-fMRI dataset from the HCP providing results also from other kernels.The authors also provided findings from simulation data.Weaknesses:(1) The authors should explain in detail how they applied cross-validation across the dataset for both optimization of parameters, and also for cross-validation of the models to predict individual traits.

Indeed, there were details about the cross-validation for hyperparameter tuning and prediction missing. This problem was also raised by Reviewer #2. We have now rephrased this section in 4.4 and added details: ll. 804-813:

“We used k-fold nested cross-validation (CV) to select and evaluate the models. We used 10 folds for both the outer loop (used to train and test the model) and the inner loop (used to select the optimal hyperparameters) such that 90% were used for training and 10% for testing. The optimal hyperparameters λ (and τ in the case of the Gaussian kernels) were selected using grid-search from the vectors λ=[0.0001,0.001,0.01,0.1,0.3,0.5,0.7,0.9,1] and τ=[1/5,1/3,1/2,1,2,3,5]. In both the outer and the inner loop, we accounted for family structure in the HCP dataset so that subjects from the same family were never split across folds (Winkler et al., 2015). Within the CV, we regressed out sex and head motion confounds, i.e., we estimated the regression coefficients for the confounds on the training set and applied them to the test set (Snoek et al., 2019).“ and ll. 818-820: “We generated the 100 random repetitions of the 10 outer CV folds once, and then used them for training and prediction of all methods, so that all methods were fit to the same partitions.”

(2) They discussed throughout the paper that their proposed (HMM+Fisher) kernel approach outperformed dynamic functional connectivity (dFC). However, they compared the proposed methodology with just static FC.

We would like to clarify that the HMM is itself a method for estimating dynamic (or time-varying) FC, just like the sliding window approach, see also Vidaurre, 2024 (https://direct.mit.edu/imag/article/doi/10.1162/imag_a_00363/124983) for an overview of terminology.

See also our response to Q3.

(3) If the authors wanted to claim that their methodology is better than dFC, then they have to demonstrate results based on dFC with the trivial sliding window approach.

We would like to be clear that we do not claim in the manuscript that our method outperforms other dynamic functional connectivity (dFC) approaches, such as sliding window FC. We have now made changes to the manuscript to make this clearer.

First, we have clarified our use of the term “brain dynamics” to signify “time-varying amplitude and functional connectivity patterns” in this context, as Reviewer #2 raised the point that the former term is ambiguous (ll.33-35: “One way of describing brain dynamics are state-space models, which allow capturing recurring patterns of activity and functional connectivity (FC) across the whole brain.”).

Second, our focus is on our method being a way of using dFC for predictive modelling, since there currently is no widely accepted way of doing this. One reason why dFC is not usually considered in prediction studies is that it is mathematically not trivial how to use the parameters from estimators of dynamic FC for a prediction. This includes the sliding window approach. We do not aim at comparing across different dFC estimators in this paper. To make these points clearer, we have revised the introduction to now say:

Ll. 39-50:

“One reason why brain dynamics are not usually considered in this context pertains to their representation: They are represented using models of varying complexity that are estimated from modalities such as functional MRI or MEG. Although there exists a variety of methods for estimating time-varying or dynamic FC (Lurie et al., 2019), like the commonly used sliding-window approach, there is currently no widely accepted way of using them for prediction problems. This is because these models are usually parametrised by a high number of parameters with complex mathematical relationships between the parameters that reflect the model assumptions. How to leverage these parameters for prediction is currently an open question.

We here propose the Fisher kernel for predicting individual traits from brain dynamics, using information from generative models that do not assume any knowledge of task timings. We focus on models of brain dynamics that capture within-session changes in functional connectivity and amplitude from fMRI scans, in this case acquired during wakeful rest, and how the parameters from these models can be used to predict behavioural variables or traits. In particular, we use the Hidden Markov Model (HMM), which is a probabilistic generative model of time-varying amplitude and functional connectivity (FC) dynamics (Vidaurre et al., 2017).”

**Reviewer #2 (Public Review):**
Summary:The manuscript presents a valuable investigation into the use of Fisher Kernels for extracting representations from temporal models of brain activity, with the aim of improving regression and classification applications. The authors provide solid evidence through extensive benchmarks and simulations that demonstrate the potential of Fisher Kernels to enhance the accuracy and robustness of regression and classification performance in the context of functional magnetic resonance imaging (fMRI) data. This is an important achievement for the neuroimaging community interested in predictive modeling from brain dynamics and, in particular, state-space models.Strengths:(1) The study's main contribution is the innovative application of Fisher Kernels to temporal brain activity models, which represents a valuable advancement in the field of human cognitive neuroimaging.(2) The evidence presented is solid, supported by extensive benchmarks that showcase the method's effectiveness in various scenarios.(3) Model inspection and simulations provide important insights into the nature of the signal picked up by the method, highlighting the importance of state rather than transition probabilities.(4) The documentation and description of the methods are solid including sufficient mathematical details and availability of source code, ensuring that the study can be replicated and extended by other researchers.Weaknesses:(1) The generalizability of the findings is currently limited to the young and healthy population represented in the Human Connectome Project (HCP) dataset. The potential of the method for other populations and modalities remains to be investigated.

As suggested by the reviewer, we have added a limitations paragraph and included a statement about the dataset: Ll. 477-481: “The fMRI dataset we used (HCP 1200 Young Adult) is a large sample taken from a healthy, young population, and it remains to be shown how our findings generalise to other datasets, e.g. other modalities such as EEG/MEG, clinical data, older populations, different data quality, or smaller sample sizes both in terms of the number of participants and the scanning duration”.

We would like to emphasise that this is a methodological contribution, rather than a basic science investigation about cognition and brain-behaviour associations. Therefore, the method would be equally usable on different populations, even if the results vary.

(2) The possibility of positivity bias in the HMM, due to the use of a population model before cross-validation, needs to be addressed to confirm the robustness of the results.

As pointed out by both Reviewers #2 and #3, we did not separate subjects into training and test set before fitting the HMM. To address this issue, we have now repeated the predictions for HMMs fit only to the training subjects. We show that this has no effect on the results. Since this question has consequences for the Fisher kernel, we have also added simulations showing how the different kernels react to increasing heterogeneity between training and test set. These new results are added as results section 2.4 (ll. 376-423).

(3) The statistical significance testing might be compromised by incorrect assumptions about the independence between cross-validation distributions, which warrants further examination or clearer documentation.

We have now replaced the significance testing with repeated k-fold cross-validated corrected tests. Note that this required re-running the models to be able to test differences in accuracies on the level of individual folds, resulting in different plots throughout the manuscript and different statistical results. This does not, however, change the main conclusions of our manuscript.

(4) The inclusion of the R^2 score, sensitive to scale, would provide a more comprehensive understanding of the method's performance, as the Pearson correlation coefficient alone is not standard in machine learning and may not be sufficient (even if it is common practice in applied machine learning studies in human neuroimaging).

We have now added the coefficient of determination to the results figures.

(5) The process for hyperparameter tuning is not clearly documented in the methods section, both for kernel methods and the elastic net.

As mentioned above in the response to Reviewer #1, we have now added details about hyperparameter tuning for the kernel methods and the non-kernelised static FC regression models (see also Reviewer #1 comment 1): Ll.804-813: “We used k-fold nested cross-validation (CV) to select and evaluate the models. We used 10 folds for both the outer loop (used to train and test the model) and the inner loop (used to select the optimal hyperparameters) such that 90% were used for training and 10% for testing. The optimal hyperparameters (and in the case of the Gaussian kernels) were selected using grid-search from the vectors λ=[0.0001,0.001,0.01,0.1,0.3,0.5,0.7,0.9,1] and τ=[1/5,1/3,1/2,1,2,3,5]. In both the outer and the inner loop, we accounted for family structure in the HCP dataset so that subjects from the same family were never split across folds (Winkler et al., 2015). Within the CV, we regressed out sex and head motion confounds, i.e., we estimated the regression coefficients for the confounds on the training set and applied them to the test set (Snoek et al., 2019).” and ll. 818-820: “We generated the 100 random repetitions of the 10 outer CV folds once, and then used them for training and prediction of all methods, so that all methods were fit to the same partitions.”, as well as ll.913-917: “All time-averaged FC models are fitted using the same (nested) cross-validation strategy as described above (10-fold CV using the outer loop for model evaluation and the inner loop for model selection using grid-search for hyperparameter tuning, accounting for family structure in the dataset, and repeated 100 times with randomised folds).”

(6) For the time-averaged benchmarks, a comparison with kernel methods using metrics defined on the Riemannian SPD manifold, such as employing the Frobenius norm of the logarithm map within a Gaussian kernel, would strengthen the analysis, cf. Jayasumana (https://arxiv.org/abs/1412.4172) Table 1, log-euclidean metric.

We have now added the log-Euclidean Gaussian kernel proposed by the reviewer to the model comparisons. The additional model does not change our conclusions.

(7) A more nuanced and explicit discussion of the limitations, including the reliance on HCP data, lack of clinical focus, and the context of tasks for which performance is expected to be on the low end (e.g. cognitive scores), is crucial for framing the findings within the appropriate context.

We have now revised the discussion section and added an explicit limitations paragraph: Ll. 475-484:

“We here aimed to show the potential of the HMM-Fisher kernel approach to leverage information from patterns of brain dynamics to predict individual traits in an example fMRI dataset as well as simulated data. The fMRI dataset we used (HCP 1200 Young Adult) is a large sample taken from a healthy, young population, and it remains to be shown how the exhibited performance generalises to other datasets, e.g. other modalities such as EEG/MEG, clinical data, older populations, different data quality, or smaller sample sizes both in terms of the number of participants and the scanning duration. Additionally, we only tested our approach for the prediction of a specific set of demographic items and cognitive scores; it may be interesting to test the framework in also on clinical variables, such as the presence of a disease or the response to pharmacological treatment.”

(8) While further benchmarks could enhance the study, the authors should provide a critical appraisal of the current findings and outline directions for future research, considering the scope and budget constraints of the work.

In addition to the new limitations paragraph (see previous comment), we have now rephrased our interpretation of the results and extended the outlook paragraph: Ll. 485-507:

“There is growing interest in combining different data types or modalities, such as structural, static, and dynamic measures, to predict phenotypes (Engemann et al., 2020; Schouten et al., 2016). While directly combining the features from each modality can be problematic, modality-specific kernels, such as the Fisher kernel for time-varying amplitude and/or FC, can be easily combined using approaches such as stacking (Breiman, 1996) or Multi Kernel Learning (MKL) (Gönen & Alpaydın, 2011). MKL can improve prediction accuracy of multimodal studies (Vaghari et al., 2022), and stacking has recently been shown to be a useful framework for combining static and time-varying FC predictions (Griffin et al., 2024). A detailed comparison of different multimodal prediction strategies including kernels for time-varying amplitude/FC may may be the focus of future work.

In a clinical context, while there are nowadays highly accurate biomarkers and prognostics for many diseases, others, such as psychiatric diseases, remain poorly understood, diagnosed, and treated. Here, improving the description of individual variability in brain measures may have potential benefits for a variety of clinical goals, e.g., to diagnose or predict individual patients’ outcomes, find biomarkers, or to deepen our understanding of changes in the brain related to treatment responses like drugs or non-pharmacological therapies (Marquand et al., 2016; Stephan et al., 2017; Wen et al., 2022; Wolfers et al., 2015). However, the focus so far has mostly been on static or structural information, leaving the potentially crucial information from brain dynamics untapped. Our proposed approach provides one avenue of addressing this by leveraging individual patterns of time-varying amplitude and FC, and it can be flexibly modified or extended to include, e.g., information about temporally recurring frequency patterns (Vidaurre et al., 2016).”

**Reviewer #3 (Public Review):**
Summary:In this work, the authors use a Hidden Markov Model (HMM) to describe dynamic connectivity and amplitude patterns in fMRI data, and propose to integrate these features with the Fisher Kernel to improve the prediction of individual traits. The approach is tested using a large sample of healthy young adults from the Human Connectome Project. The HMM-Fisher Kernel approach was shown to achieve higher prediction accuracy with lower variance on many individual traits compared to alternate kernels and measures of static connectivity. As an additional finding, the authors demonstrate that parameters of the HMM state matrix may be more informative in predicting behavioral/cognitive variables in this data compared to state-transition probabilities.Strengths:- Overall, this work helps to address the timely challenge of how to leverage high-dimensional dynamic features to describe brain activity in individuals.- The idea to use a Fisher Kernel seems novel and suitable in this context.- Detailed comparisons are carried out across the set of individual traits, as well as across models with alternate kernels and features.- The paper is well-written and clear, and the analysis is thorough.Potential weaknesses:- One conclusion of the paper is that the Fisher Kernel "predicts more accurately than other methods" (Section 2.1 heading). I was not certain this conclusion is fully justified by the data presented, as it appears that certain individual traits may be better predicted by other approaches (e.g., as shown in Figure 3) and I found it hard to tell if certain pairwise comparisons were performed -- was the linear Fisher Kernel significantly better than the linear Naive normalized kernel, for example?

We have revised the abstract and the discussion to state the results more appropriately. For instance, we changed the relevant section in the abstract to (ll. 24-26):

“We show here, in fMRI data, that the HMM-Fisher kernel approach is accurate and reliable. We compare the Fisher kernel to other prediction methods, both time-varying and time-averaged functional connectivity-based models.”,

and in the discussion, removing the sentence

“resulting in better generalisability and interpretability compared to other methods”,

and adding (given the revised statistical results) ll. 435-436:

“though most comparisons were not statistically significant given the narrow margin for improvements.”

In conjunction with the new statistical approach (see Reviewer #2, comment 3), we have now streamlined the comparisons. We explained which comparisons were performed in the methods ll.880-890:

“For the main results, we separately compare the linear Fisher kernel to the other linear kernels, and the Gaussian Fisher kernel to the other Gaussian kernels, as well as to each other. We also compare the linear Fisher kernel to all time-averaged methods. Finally, to test for the effect of tangent space projection for the time-averaged FC prediction, we also compare the Ridge regression model to the Ridge Regression in Riemannian space. To test for effects of removing sets of features, we use the approach described above to compare the kernels constructed from the full feature sets to their versions where features were removed or reduced. Finally, to test for effects of training the HMM either on all subjects or only on the subjects that were later used as training set, we compare each kernel to the corresponding kernel constructed from HMM parameters, where training and test set were kept separate.“

Model performance evaluation is done on the level of all predictions (i.e., across target variables, CV folds, and CV iterations) rather than for each of the target variables separately. That means different best-performing methods depending on the target variables are to be expected.

- While 10-fold cross-validation is used for behavioral prediction, it appears that data from the entire set of subjects is concatenated to produce the initial group-level HMM estimates (which are then customized to individuals). I wonder if this procedure could introduce some shared information between CV training and test sets. This may be a minor issue when comparing the HMM-based models to one another, but it may be more important when comparing with other models such as those based on time-averaged connectivity, which are calculated separately for train/test partitions (if I understood correctly).

The lack of separation between training and test set before fitting the HMM was also pointed out by Reviewer #2. We are addressing this issue in the new Results section 2.4 (see also our response to Reviewer #2, comment 2).

**Recommendations for the authors:**
The individual public reviews all indicate the merits of the study, however, they also highlight relatively consistent questions or issues that ought to be addressed. Most significantly, the authors ought to provide greater clarity surrounding the use of the cross-validation procedures they employ, and the use of a common atlas derived outside the cross-validation loop. Also, the authors should ensure that the statistical testing procedures they employ accommodate the dependencies induced between folds by the cross-validation procedure and give care to ensuring that the conclusions they make are fully supported by the data and statistical tests they present.
**Reviewer #1 (Recommendations For The Authors):**
Overall, the study is interesting but demands further improvements. Below, I summarize my comments:(1) The authors should explain in detail how they applied cross-validation across the dataset for both optimization of parameters, and also for cross-validation of the models to predict individual traits.How did you split the dataset for both parameters optimization, and for the CV of the prediction of behavioral traits?A review and a summary of various CVs that have been applied on the same dataset should be applied.

We apologise for the oversight and have now added more details to the CV section of the methods, see our response to Reviewer #1 comment 1:

In ll. 804-813:

“We used k-fold nested cross-validation (CV) to select and evaluate the models. We used 10 folds for both the outer loop (used to train and test the model) and the inner loop (used to select the optimal hyperparameters) such that 90% were used for training and 10% for testing. The optimal hyperparameters (and in the case of the Gaussian kernels) were selected using grid-search from the vectors λ=[0.0001,0.001,0.01,0.1,0.3,0.5,0.7,0.9,1] and τ=[1/5,1/3,1/2,1,2,3,5]. In both the outer and the inner loop, we accounted for family structure in the HCP dataset so that subjects from the same family were never split across folds (Winkler et al., 2015). Within the CV, we regressed out sex and head motion confounds, i.e., we estimated the regression coefficients for the confounds on the training set and applied them to the test set (Snoek et al., 2019).“ and ll. 818-820: “We generated the 100 random repetitions of the 10 outer CV folds once, and then used them for training and prediction of all methods, so that all methods were fit to the same partitions.”

(2) The authors should explain in more detail how they applied ICA-based parcellation at the group-level.A. Did you apply it across the whole group? If yes, then this is problematic since it rejects the CV approach. It should be applied within the folds.B. How did you define the representative time-source per ROI?

A: How group ICA was applied was stated in the Methods section (4.1 HCP imaging and behavioural data), ll. 543-548:

“The parcellation was estimated from the data using multi-session spatial ICA on the temporally concatenated data from all subjects.”

We have now added a disclaimer about the divide between training and test set:

“Note that this means that there is no strict divide between the subjects used for training and the subjects for testing the later predictive models, so that there is potential for leakage of information between training and test set. However, since this step does not concern the target variable, but only the preprocessing of the predictors, the effect can be expected to be minimal (Rosenblatt et al., 2024).”

We understand that in order to make sure we avoid data leakage, it would be desirable to estimate and apply group ICA separately for the folds, but the computational load of this would be well beyond the constraints of this particular work, where we have instead used the parcellation provided by the HCP consortium.

B: This was also stated in 4.1, ll. 554-559: “Timecourses were extracted using dual regression (Beckmann et al., 2009), where group-level components are regressed onto each subject’s fMRI data to obtain subject-specific versions of the parcels and their timecourses. We normalised the timecourses of each subject to ensure that the model of brain dynamics and, crucially, the kernels were not driven by (averaged) amplitude and variance differences between subjects.”

(3) The authors discussed throughout the paper that their proposed (HMM+Fisher) kernel approach outperformed dynamic functional connectivity (dFC). However, they compared the proposed methodology with just static FC.A. The authors didn't explain how static and dFC have been applied.B. If the authors wanted to claim that their methodology is better than dFC, then they have to demonstrate results based on dFC with the trivial sliding window approach.C. Moreover, the static FC networks have been constructed by concatenating time samples that belong to the same state across the time course of resting-state activity.So, it's HMM-informed static FC analysis, which is problematic since it's derived from HMM applied over the brain dynamics.I don't agree that connectivity is derived exclusively from the clustering of human brain dynamics!D. A static approach of using the whole time course, and a dFC following the trivial sliding-window approach should be adopted and presented for comparison with (HMM+Fisher) kernel.

We do not intend to claim our manuscript that our method outperforms other methods for doing dynamic FC. Indeed, we would like to be clear that the HMM itself is a method for capturing dynamic FC. Please see our responses to public review comments 2 and 3 by reviewer #1, copied below, which is intended to clear up this misunderstanding:

We would like to clarify that the HMM is itself a method for estimating dynamic (or time-varying) FC, just like the sliding window approach, see also Vidaurre, 2024 (https://direct.mit.edu/imag/article/doi/10.1162/imag_a_00363/124983) for an overview of terminology.

We would like to be clear that we do not claim in the manuscript that our method outperforms other dynamic functional connectivity (dFC) approaches, such as sliding window FC. We have now made changes to the manuscript to make this clearer.

First, we have clarified our use of the term “brain dynamics” to signify “time-varying amplitude and functional connectivity patterns” in this context, as Reviewer #2 raised the point that the former term is ambiguous.

Second, our focus is on our method being a way of using dFC for predictive modelling, since there currently is no widely accepted way of doing this. One reason why dFC is not usually considered in prediction studies is that it is mathematically not trivial how to use the parameters from estimators of dynamic FC for a prediction. This includes the sliding window approach. We do not aim at comparing across different dFC estimators in this paper. To make these points clearer, we have revised the introduction to now say:

Ll. 39-50:

“One reason why brain dynamics are not usually considered in this context pertains to their representation: They are represented using models of varying complexity that are estimated from modalities such as functional MRI or MEG. Although there exists a variety of methods for estimating time-varying or dynamic FC (Lurie et al., 2019), like the commonly used sliding-window approach, there is currently no widely accepted way of using them for prediction problems. This is because these models are usually parametrised by a high number of parameters with complex mathematical relationships between the parameters that reflect the model assumptions. How to leverage these parameters for prediction is currently an open question.

We here propose the Fisher kernel for predicting individual traits from brain dynamics, using information from generative models that do not assume any knowledge of task timings. We focus on models of brain dynamics that capture within-session changes in functional connectivity and amplitude from fMRI scans, in this case acquired during wakeful rest, and how the parameters from these models can be used to predict behavioural variables or traits. In particular, we use the Hidden Markov Model (HMM), which is a probabilistic generative model of time-varying amplitude and functional connectivity (FC) dynamics (Vidaurre et al., 2017).”

To the additional points raised here:

A: How static and dynamic FC have been estimated is explicitly stated in the relevant Methods sections 4.2 (The Hidden Markov Model), which explains the details of using the HMM to estimate dynamic functional connectivity; and 4.5 (Regression models based on time-averaged FC features), which explains how static FC was computed.

B: We are not making this claim. We have now modified the Introduction to avoid further misunderstandings, as per ll. 33-36: “One way of describing brain dynamics are state-space models, which allow capturing recurring patterns of activity and functional connectivity (FC) across the whole brain.”

C: This is not how static FC networks were constructed; we apologise for the confusion. We also do not perform any kind of clustering. The only “HMM-informed static FC analysis” is the static FC KL divergence model to allow for a more direct comparison with the time-varying FC KL divergence model, but we have included several other static FC models (log-Euclidean, Ridge regression, Ridge regression Riem., Elastic Net, Elastic Net Riem., and Selected Edges), which do not use HMMs. This is explained in Methods section 4.5.

D: As explained above, we have included four (five in the revised manuscript) static approaches using the whole time course, and we do not claim that our method outperforms other dynamic FC models. We also disagree that using the sliding window approach for predictive modelling is trivial, as explained in the introduction of the manuscript and under public review comment 3.

(4) Did you correct for multiple comparisons across the various statistical tests?

All statistical comparisons have been corrected for multiple comparisons. Please find the relevant text in Methods section 4.4.1.

(5) Do we expect that behavioral traits are encapsulated in resting-state human brain dynamics, and on which brain areas mostly? Please, elaborate on this.

While this is certainly an interesting question, our paper is a methodological contribution about how to predict from models of brain dynamics, rather than a basic science study about the relation between resting-state brain dynamics and behaviour. The biological aspects and interpretation of the specific brain-behaviour associations are a secondary point and out of scope for this paper. Our approach uses whole-brain dynamics, which does not require selecting brain areas of interest.

**Reviewer #2 (Recommendations For The Authors):**
Beyond the general principles included in the public review, here are a few additional pointers to minor issues that I would wish to see addressed.Introduction:- The term "brain dynamics" encompasses a broad spectrum of phenomena, not limited to those captured by state-space models. It includes various measures such as time-averaged connectivity and mean EEG power within specific frequency bands. To ensure clarity and relevance for a diverse readership, it would be beneficial to adopt a more inclusive and balanced approach to the terminology used.

The reviewer rightly points out the ambiguity of the term “brain dynamics”, which we use in the interest of readability. The HMM is one of several possible descriptions of brain dynamics. We have now included a statement early in the introduction to narrow this down:

Ll. 32-35:

“… the patterns in which brain activity unfolds over time, i.e., brain dynamics. One way of describing brain dynamics are state-space models, which allow capturing recurring patterns of activity and functional connectivity (FC) across the whole brain.”

And ll. 503-507:

“Our proposed approach provides one avenue of addressing this by leveraging individual patterns of time-varying amplitude and FC, as one of many possible descriptions of brain dynamics, and it can be flexibly modified or extended to include, e.g., information about temporally recurring frequency patterns (Vidaurre et al., 2016).”

Figures:- The font sizes across the figures, particularly in subpanels 2B and 2C, are quite small and may challenge readability. It is advisable to standardize the font sizes throughout all figures to enhance legibility.

We have slightly increased the overall font sizes, while we are generally following figure recommendations set out by Nature. The font sizes are the same throughout the figures.

- When presenting performance comparisons, a horizontal layout is often more intuitive for readers, as it aligns with the natural left-to-right reading direction. This is not just a personal preference; it is supported by visualization best practices as outlined in resources like the NVS Cheat Sheet (https://github.com/GraphicsPrinciples/CheatSheet/blob/master/NVSCheatSheet.pdf) and Kieran Healy's book (https://socviz.co/lookatdata.html).

We have changed all figures to use horizontal layout, hoping that this will ease visual comparison between the different models.

- In the kernel density estimation (KDE) and violin plot representations, it appears that the data displays may be truncated. It is crucial to indicate where the data distribution ends. Overplotting individual data points could provide additional clarity.

To avoid confusion about the data distribution in the violin plots, we have now overlaid scatter plots, as suggested by the reviewer. Overlaying the fold-level accuracies was not feasible (since this would result in ~1.5 million transparent points for a single figure), so we instead show the accuracies averaged over folds but separate for target variables and CV iterations. Only the newly added coefficient of determination plots had to be truncated, which we have noted in the figure legend.

- Figure 3 could inadvertently suggest that time-varying features correspond to panel A and time-averaged features to panel B. To avoid confusion, consider reorganizing the labels at the bottom into two rows for clearer attribution.

We have changed the layout of the time-varying and time-averaged labels in the new version of the plots to avoid this issue.

Discussion:- The discussion on multimodal modeling might give the impression that it is more effective with multiple kernel learning (MKL) than with other methods. To present a more balanced view, it would be appropriate to rephrase this section. For instance, stacking, examples of which are cited in the same paragraph, has been successfully applied in practice. The text could be adjusted to reflect that Fisher Kernels via MKL adds to the array of viable options for multimodal modeling. As a side thought: additionally, a well-designed comparison between MKL and stacking methods, conducted by experts in each domain, could greatly benefit the field. In certain scenarios, it might even be demonstrated that the two approaches converge, such as when using linear kernels.

We would like to thank the reviewer for the suggestion about the discussion concerning multimodal modelling. We agree that there are other relevant methods that may lead to interesting future work and have now included stacking and refined the section: ll. 487-494:

“While directly combining the features from each modality can be problematic, modality-specific kernels, such as the Fisher kernel for time-varying amplitude and/or FC, can be easily combined using approaches such as stacking (Breiman, 1996) or Multi Kernel Learning (MKL) (Gönen & Alpaydın, 2011). MKL can improve prediction accuracy of multimodal studies (Vaghari et al., 2022), and stacking has recently been shown to be a useful framework for combining static and time-varying FC predictions (Griffin et al., 2024). A detailed comparison of different multimodal prediction strategies including kernels for time-varying amplitude/FC may be the focus of future work.”

- The potential clinical applications of brain dynamics extend beyond diagnosis and individual outcome prediction. They play a significant role in the context of biomarkers, including pharmacodynamics, prognostic assessments, responder analysis, and other uses. The current discussion might be misinterpreted as being specific to hidden Markov model (HMM) approaches. For diagnostic purposes, where clinical assessment or established biomarkers are already available, the need for new models may be less pressing. It would be advantageous to reframe the discussion to emphasize the potential for gaining deeper insights into changes in brain activity that could indicate therapeutic effects or improvements not captured by structural brain measures. However, this forward-looking perspective is not the focus of the current work. A nuanced revision of this section is recommended to better reflect the breadth of applications.

We appreciate the reviewer’s thoughtful suggestions regarding the discussion of potential clinical applications. We have included the suggestions and refined this section of the discussion: Ll. 495-507:

“In a clinical context, while there are nowadays highly accurate biomarkers and prognostics for many diseases, others, such as psychiatric diseases, remain poorly understood, diagnosed, and treated. Here, improving the description of individual variability in brain measures may have potential benefits for a variety of clinical goals, e.g., to diagnose or predict individual patients’ outcomes, find biomarkers, or to deepen our understanding of changes in the brain related to treatment responses like drugs or non-pharmacological therapies (Marquand et al., 2016; Stephan et al., 2017; Wen et al., 2022; Wolfers et al., 2015). However, the focus so far has mostly been on static or structural information, leaving the potentially crucial information from brain dynamics untapped. Our proposed approach provides one avenue of addressing this by leveraging individual patterns of time-varying amplitude and FC, and it can be flexibly modified or extended to include, e.g., information about temporally recurring frequency patterns (Vidaurre et al., 2016).”

**Reviewer #3 (Recommendations For The Authors):**
- I wondered if the authors could provide, within the Introduction, an intuitive description for how the Fisher Kernel "preserves the structure of the underlying model of brain dynamics" / "preserves the mathematical structure of the underlying HMM"? Providing more background may help to motivate this study to a general audience.

We agree that this would be helpful and have now added this to the introduction: Ll.61-67:

“Mathematically, the HMM parameters lie on a Riemannian manifold (the structure). This defines, for instance, the relation between parameters, such as: how changing one parameter, like the probabilities of transitioning from one state to another, would affect the fitting of other parameters, like the states’ FC. It also defines the relative importance of each parameter; for example, how a change of 0.1 in the transition probabilities would not be the same as a change of 0.1 in one edge of the states’ FC matrices.”

To communicate the intuition behind the concept, the idea was also illustrated in Figure 1, panel 4 by showing Euclidean distances as straight lines through a curved surface (4a, Naïve kernel), as opposed to the tangent space projection onto the curved manifold (4b, Fisher kernel).

- Some clarifications regarding Figure 2a would be helpful. Was the linear Fisher Kernel significantly better than the linear Naive normalized kernel? I couldn't find whether this comparison was carried out. Apologies if I have missed it in the text. For some of the brackets indicating pairwise tests and their significance values, the start/endpoints of the bracket fall between two violins; in this case, were the results of the linear and Gaussian Fisher Kernels pooled together for this comparison?

We have now streamlined the statistical comparisons and avoided plotting brackets falling between two violin plots. The comparisons that were carried out are stated in the methods section 4.4.1. Please see also our response to above to Reviewer #3 public review, potential weaknesses, point 1, relevant point copied below:

In conjunction with the new statistical approach (see Reviewer #2, comment 3), we have now streamlined the comparisons. We explained which comparisons were performed in the methods ll.880-890:

“For the main results, we separately compare the linear Fisher kernel to the other linear kernels, and the Gaussian Fisher kernel to the other Gaussian kernels, as well as to each other. We also compare the linear Fisher kernel to all time-averaged methods. Finally, to test for the effect of tangent space projection for the time-averaged FC prediction, we also compare the Ridge regression model to the Ridge Regression in Riemannian space. To test for effects of removing sets of features, we use the approach described above to compare the kernels constructed from the full feature sets to their versions where features were removed or reduced. Finally, to test for effects of training the HMM either on all subjects or only on the subjects that were later used as training set, we compare each kernel to the corresponding kernel constructed from HMM parameters, where training and test set were kept separate”.

- The authors may wish to include, in the Discussion, some remarks on the use of all subjects in fitting the group-level HMM and the implications for the cross-validation performance, and/or try some analysis to ensure that the effect is minor.

As suggested by reviewers #2 and #3, we have now performed the suggested analysis and show that fitting the group-level HMM to all subjects compared to only to the training subjects has no effect on the results. Please see our response to Reviewer #2, public review, comment 2.

- The decision to use k=6 states was made here, and I wondered if the authors may include some support for this choice (e.g., based on findings from prior studies)?

We have now refined and extended our explanation and rationale behind the number of states: Ll. 586-594: “The number of states can be understood as the level of detail or granularity with which we describe the spatiotemporal patterns in the data, akin to a dimensionality reduction, where a small number of states will lead to a very general, coarse description and a large number of states will lead to a very detailed, fine-grained description. Here, we chose a small number of states, K=6, to ensure that the group-level HMM states are general enough to be found in all subjects, since a larger number of states increases the chances of certain states being present only in a subset of subjects. The exact number of states is less relevant in this context, since the same HMM estimation is used for all kernels.”

- (minor) Abstract: "structural aspects" - do you mean structural connectivity?

With “structural aspects”, we refer to the various measures of brain structure that are used in predictive modelling. We have now specified: Ll. 14-15: “structural aspects, such as structural connectivity or cortical thickness”.